# Dynamic regulation of GDP binding to G proteins revealed by magnetic field-dependent NMR relaxation analyses

Yuki Toyama[1,2], Hanaho Kano[1], Yoko Mase[1], Mariko Yokogawa[1,†], Masanori Osawa[1,†] & Ichio Shimada[1]

Heterotrimeric guanine-nucleotide-binding proteins (G proteins) serve as molecular switches in signalling pathways, by coupling the activation of cell surface receptors to intracellular responses. Mutations in the G protein α-subunit (Gα) that accelerate guanosine diphosphate (GDP) dissociation cause hyperactivation of the downstream effector proteins, leading to oncogenesis. However, the structural mechanism of the accelerated GDP dissociation has remained unclear. Here, we use magnetic field-dependent nuclear magnetic resonance relaxation analyses to investigate the structural and dynamic properties of GDP bound Gα on a microsecond timescale. We show that Gα rapidly exchanges between a ground-state conformation, which tightly binds to GDP and an excited conformation with reduced GDP affinity. The oncogenic D150N mutation accelerates GDP dissociation by shifting the equilibrium towards the excited conformation.

[1] Graduate School of Pharmaceutical Sciences, The University of Tokyo, Hongo, Bunkyo-ku, Tokyo 113-0033, Japan. [2] Japan Biological Informatics Consortium (JBiC), Aomi, Koto-ku, Tokyo 135-0064, Japan. † Present address: Keio University Faculty of Pharmacy, Shibakoen, Minato-ku, Tokyo 105-8512, Japan. Correspondence and requests for materials should be addressed to I.S. (email: shimada@iw-nmr.f.u-tokyo.ac.jp).

Heterotrimeric guanine-nucleotide-binding proteins (G proteins) serve as molecular switches in signalling pathways, by coupling the activation of G protein-coupled receptors (GPCRs) at the cell surface to intracellular responses[1,2]. The signalling pathways, involving G proteins, play central roles in a wide variety of biological processes, including cellular division, immune responses and sensory functions. Heterotrimeric G proteins are composed of three subunits, an α-subunit (Gα), a β-subunit (Gβ) and a γ-subunit (Gγ). The latter two subunits are tightly associated and form a Gβγ dimer under physiological conditions. In the resting state, the G protein forms a heterotrimer, consisting of a guanosine diphosphate (GDP)-bound form of Gα (Gα-GDP) and Gβγ. Ligand binding to GPCRs induces conformational changes of Gα-GDP, promoting the dissociation of GDP. Guanosine triphosphate (GTP) spontaneously binds to Gα, resulting in the dissociation of the GTP-bound form of Gα (Gα-GTP) and Gβγ. Subsequently, both Gα-GTP and Gβγ bind to the effector proteins and regulate their functions, leading to cellular responses. The G protein returns to the resting state, following GTP hydrolysis and the re-association of Gα-GDP and Gβγ. The hydrolysis and the re-association of Gα-GDP are regulated by the binding of regulators of G protein signalling (RGS) to the G proteins, which controls both the length and strength of the effector regulations[3].

Based on primary sequence similarity, the Gα proteins can be classified into four major sub-families, i/o, s, q/11 and 12/13, and each couples with a different group of GPCRs and regulates different signalling pathways[1]. The three-dimensional structures of Gα have a conserved protein fold, which is composed of a GTPase domain and a helical domain (Supplementary Fig. 1a). The GTPase domain consists of five helices (α1–α5) surrounding six β-sheets (β1–β6), and is homologous to the Ras-like small GTPase proteins. The helical domain consists of six helices (αA–αF), and is linked to the GTPase domain by two linker regions. The bound guanine nucleotide forms interactions with residues from both of the domains, and resides in a cleft between the two domains (Supplementary Fig. 1b,c). The GTPase domain contains three flexible loops, named switches 1, 2 and 3, where significant structural changes occur between the Gα-GTP and Gα-GDP structures. These regions are involved in binding to Gβγ, the effector proteins, and RGS.

Mutations in Gα have been found in various type of cancers, and the aberrant function of Gα is considered to be strongly associated with tumour genesis[4–8]. Genome sequencing studies have revealed that the mutations in Gα_q are frequently found in some tumour types, lacking B-Raf and N-Ras mutations, indicating that mutations in Gα potentially play a significant role in cancers[4]. Among these oncogenic mutations, some mutations, such as D151N (αE helix) and R243H (α3 helix) in Gα_o, are known to promote oncogenesis by accelerating the GDP dissociation without the catalytic activity of the activated GPCR[8–10]. Since the intracellular GTP/GDP ratio is high (~10), the accelerated GDP dissociation leads to an increase in the fraction of active Gα-GTP, causing the hyperactivation of the effector proteins, which leads to oncogenesis (Fig. 1a). Along with the sequencing analyses of cancer genomes, biochemical studies have also identified mutations with the accelerated GDP dissociation[11–14]. These mutated residues do not form direct interactions with the bound GDP, suggesting that the dissociation of GDP is facilitated in allosteric manners. In some cases, the crystal structures of the mutants have been solved; however, the structural differences in the GDP-binding modes have not clearly been observed[14,15]. Therefore, the structural mechanism of the GDP dissociation and the means by which the oncogenic mutations accelerate the GDP dissociation still remain unclear.

Here, we investigate the structural and dynamic properties of Gα-GDP with solution nuclear magnetic resonance (NMR) spectroscopy. Newly established NMR methods allow us to characterize chemical exchange processes on a microsecond timescale. We characterize the conformational exchange processes of wild-type Gα-GDP and its oncogenic D150N mutant, which allow us to explain the observed different GDP affinities of both proteins. Based on these results, we propose a dynamic regulatory mechanism for GDP dissociation.

## Results

**GDP dissociates faster in the D150N mutant.** In this study, we focused on the D150N mutant of Gα_i3. The corresponding mutation has been found in Gα_t, Gα_16 and Gα_z in cancer[10,16]. The dissociation rate of GDP is accelerated in the mutants, leading to a gain-of-function in cancer cells. We purified the wild-type and D150N mutant proteins and measured their GDP dissociation rate, using [3]H-labelled GDP ([3]H-GDP) as a tracer. The dissociation rate constants of [3]H-GDP were $0.0072\,min^{-1}$ for the wild type, and $0.14\,min^{-1}$ for the D150N mutant at 20 °C. Thus, the GDP dissociation in the D150N mutant was ~20-fold faster than that in the wild type (Fig. 1b). Although the constitutive activation of the effector proteins by the D150N mutant has not been confirmed in cells, the Gα_o mutant with the GDP dissociation rate sixfold faster than that of the wild type is reported to significantly enhance the activation of the effector proteins, and promote anchorage-independent growth in human mammary epithelial cells[8,9]. Therefore, we conclude that the GDP dissociation rate in the D150N mutant is sufficiently high to promote the oncogenic signalling.

The crystal structure of wild-type Gα-GDP (ref. 17) revealed that Asp150, located on the αE helix in the helical domain, forms no direct interactions with the bound GDP, therefore, the accelerated GDP dissociation seems to result from the structural differences in Gα-GDP induced by the mutation. To gain structural insights into the mechanism of the GDP dissociation, we conducted NMR analyses for the wild type and the D150N mutant. Because of the relatively large molecular weight of Gα-GDP (41 K), we adopted selective methyl-labelling strategies and applied methyl-TROSY techniques[18,19]. We prepared a selectively labelled {u-[2]H, Ileδ1, Leuδ2, Valγ2-[[13]CH_3]} Gα-GDP sample, and compared the [1]H-[13]C heteronuclear multiple quantum (MQ) coherence (HMQC) spectra (Fig. 1c and Supplementary Fig. 2). Remarkably, the chemical shift differences between the wild type and the D150N mutant were quite small, and the weighted average chemical shift differences were <0.1 p.p.m. for all methyl groups, suggesting that the overall structure is well preserved in the mutant. Chemical shift differences larger than 0.03 p.p.m. were observed for Leu148, Val174, Val223 and Leu234, which did not form direct contacts with the bound GDP, suggesting that the interactions with the bound GDP were not significantly perturbed in the D150N mutant (Fig. 1e).

Numerous studies have shown that ligand dissociation occurs from a low-affinity structure, which is only transiently formed and is exchanging with the ground-state structure[20,21]. In support of this idea, an increase in conformational flexibility during nucleotide exchange has been noted in previous structural studies of Gα (refs 14,22). To determine whether such conformational exchange processes exist and contribute to the GDP dissociation process in Gα-GDP, we conducted MQ Carr–Purcell–Meiboom–Gill relaxation dispersion (MQ CPMG RD) experiments, to characterize the chemical exchange processes on millisecond to microsecond timescales[23] (Fig. 1d,e). In the wild type, small but significant differences in the effective relaxation rates ($R_{2,eff}$)

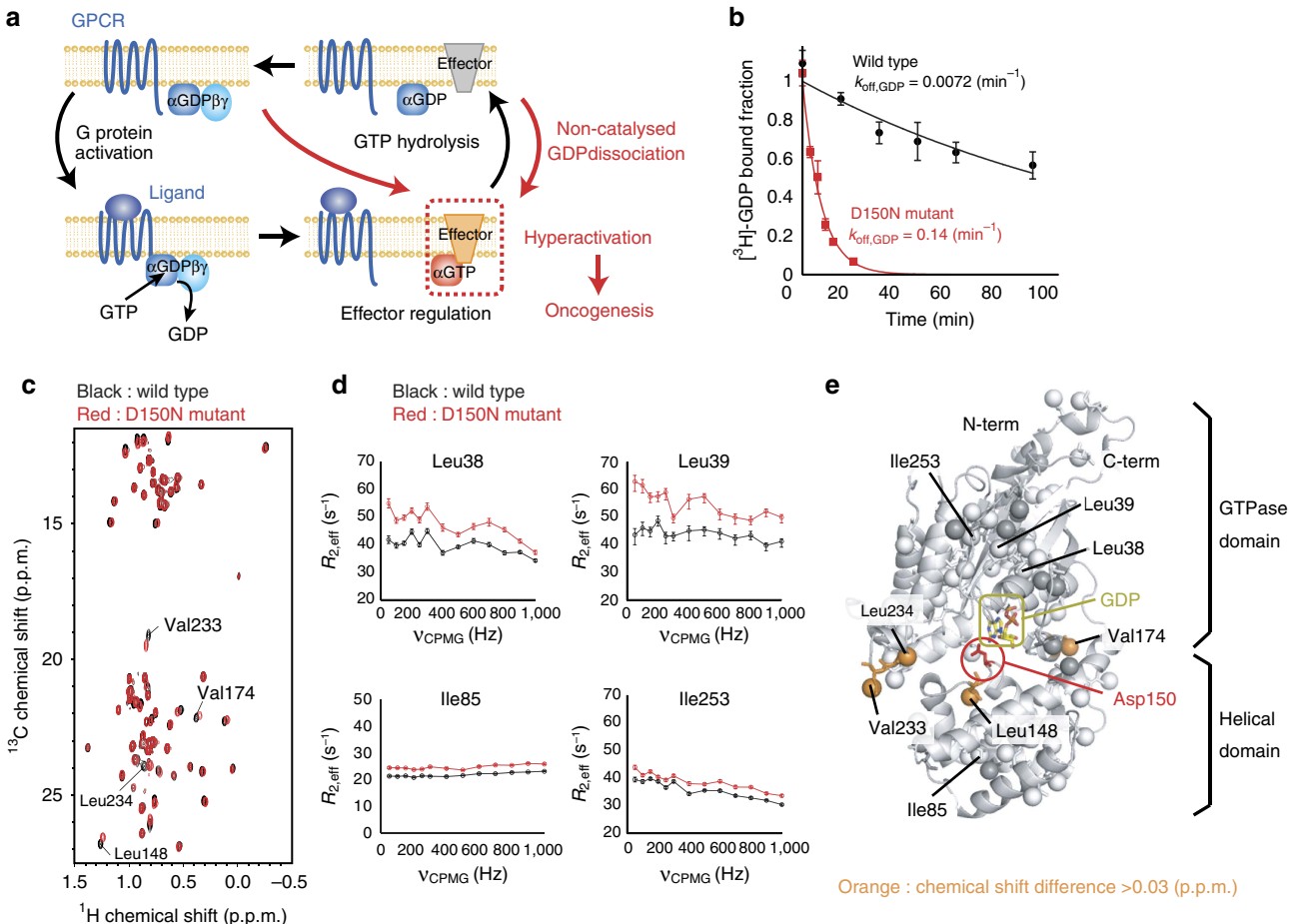

**Figure 1 | Functional and structural analyses of the wild-type and mutant Gα-GDP.** (**a**) Proposed mechanism for oncogenesis induced by the Gα mutant[9]. In cells expressing the Gα mutant that shows accelerated GDP dissociation, the GDP-GTP exchange reaction is facilitated without the activated GPCR, and the fraction of Gα-GTP is increased. The increased Gα-GTP causes hyperactivation of the effector proteins, leading to oncogenesis. (**b**) [3]H-GDP dissociation assays of the wild-type Gα and the D150N mutant. The experiments were performed at 20 °C. Each point reflects mean ± s.e. of three independent experiments. (**c**) Overlay of the [1]H-[13]C HMQC spectra of the wild type (black) and the D150N mutant (red). The methyl groups with chemical shift differences larger than 0.03 p.p.m. are shown labelled. (**d**) Results of the MQ CPMG RD experiments with the wild type (black) and the D150N mutant (red). The experiments were performed at 14.1 Tesla (600 MHz [1]H frequency). The error bars represent the experimental errors calculated using the equation (16). (**e**) Mapping of the methyl groups with chemical shift differences in the D150N mutant on the structure of Gα-GDP (PDB ID: 1GDD)[17]. Methyl groups with chemical shift differences larger than 0.03 p.p.m. are coloured orange. Methyl groups with no data are coloured grey. The methyl groups for which MQ CPMG RD results are presented (Leu38, Leu39, Ile85 and Ile253) are also shown with labels.

larger than $5.0\,\mathrm{s^{-1}}$ were observed for Leu38 and Ile253, which are located on a β1-strand and an α3-helix, respectively. The $R_{2,\mathrm{eff}}$ rates decreased linearly with increasing CPMG frequencies, and the changes were not saturated with the 1,000 Hz CPMG pulse, suggesting that the exchange processes occur on a much faster timescale than the CPMG frequencies, which occur on a microsecond timescale (an exchange rate of $10^3$–$10^4\,\mathrm{s^{-1}}$). We also conducted the MQ CPMG RD experiments with the D150N mutant, for comparison with the wild type. Notably, larger differences in the $R_{2,\mathrm{eff}}$ rates were observed for Leu38 and Leu39, located on the β1-strand, as compared with the wild type.

These results demonstrated that conformational exchange processes exist in Gα-GDP, and the exchange parameters are different between the wild type and the D150N mutant, suggesting that the exchange processes are closely related to the GDP dissociation. However, the differences in the $R_{2,\mathrm{eff}}$ rates were small and detected in only a few methyl groups, thus hampering the quantitative and comprehensive analyses of the conformational exchange processes in Gα-GDP.

**MQ relaxation analyses of Gα-GDP.** Although the MQ CPMG RD experiments are beneficial for characterizing the chemical exchange processes in high molecular weight proteins, the accessible timescales are often limited to exchange rates of $10^2$–$10^3\,\mathrm{s^{-1}}$, mainly due to the upper limit of the frequencies of the CPMG pulse trains. The faster exchange processes, with exchange rates on the order of $10^3$–$10^4\,\mathrm{s^{-1}}$, can be characterized by spin-lock-based $R_{1\mathrm{rho}}$ dispersion experiments[24]. However, when observing methyl groups, the $R_{1\mathrm{rho}}$ dispersion experiments require $^{13}\mathrm{CH^2H_2}$ labelled samples, in order to avoid artefacts derived from intra-methyl dipolar cross-correlation[25]. The $^{13}\mathrm{CH^2H_2}$ labelling results in lower sensitivities (about threefold), mainly due to the reduced initial magnetization[26]. Therefore, the applications of the $R_{1\mathrm{rho}}$ dispersion experiments have been limited to small- or medium-sized proteins, so far[25,27], and a new strategy is needed for characterizing the microsecond-order conformational exchange processes in high molecular weight proteins, such as Gα-GDP.

To overcome these difficulties, we developed an NMR method to detect and characterize the chemical exchange processes on a

microsecond timescale from the side-chain methyl $^1$H-$^{13}$C MQ relaxation rates, by utilizing the static magnetic field dependency[28–30]. Assuming a simple two-state (states A and B) chemical exchange process, the exchange contributions in the $^1$H-$^{13}$C MQ relaxation rates, $R_{MQ,ex}$ and $\Delta R_{MQ,ex}$, can be expressed by equations (1) and (2), using the exchange rate, $k_{ex}$, the populations of the two states, $p_A$, $p_B$ ($p_A > p_B$), and the $^{13}$C and $^1$H chemical shift differences given in p.p.m. units, $\Delta\varpi_C$ and $\Delta\varpi_H$ (refs 31,32).

$$R_{MQ,ex} = \frac{(\gamma_C^2 \Delta\varpi_C^2 + \gamma_H^2 \Delta\varpi_H^2)p_A p_B}{k_{ex}} \cdot B_0^2 \tag{1}$$

$$\Delta R_{MQ,ex} = \frac{4\gamma_C \Delta\varpi_C \gamma_H \Delta\varpi_H p_A p_B}{k_{ex}} \cdot B_0^2 \tag{2}$$

The $R_{MQ,ex}$ and $\Delta R_{MQ,ex}$ rates can be extracted, using (i) the $R_{MQ}$ and $\Delta R_{MQ}$ rates measured at different static magnetic fields, (ii) the chemical shift anisotropy (CSA) values, $\Delta\sigma_C$ and $\Delta\sigma_H$ and (iii) the product of the order parameter and the rotational correlation time, $S^2_{axis}\tau_C$, as summarized in Fig. 2a. We verified that the $R_{MQ,ex}$ and $\Delta R_{MQ,ex}$ rates are highly sensitive to the chemical exchange processes on a microsecond timescale, and that accurate $R_{MQ,ex}$ and $\Delta R_{MQ,ex}$ rates can be obtained by the magnetic field-dependent MQ relaxation analyses in high molecular weight proteins (Supplementary Figs 3–6).

To examine the conformational exchange processes in Gα-GDP, we applied the magnetic field-dependent MQ relaxation analyses to Gα-GDP. Figure 2b shows the plot of the $R_{MQ}$ rates of Leu38, Leu39, Ile85 and Ile253, as a function of the square of $B_0$. Significantly large magnetic field-dependent changes were observed for Leu38, Leu39 and Ile253, reflecting the $R_{MQ,ex}$ rates over $20\,s^{-1}$ at 14.1 Tesla (600 MHz $^1$H frequency). These results were in contrast to the results from the MQ CPMG RD analyses, where the observed differences in the $R_{2,eff}$ rates were as large as $5.0\,s^{-1}$, demonstrating the robustness of the method. The plots of the $R_{MQ,ex}$ and $\Delta R_{MQ,ex}$ rates at 14.1 Tesla are shown in Fig. 2c, and the methyl groups with significant $R_{MQ,ex}$ and $\Delta R_{MQ,ex}$ rates were mapped on the structure of Gα-GDP (Fig. 2d). These methyl groups were located on the β1-strand (Leu38, Leu39), α1-helix (Ile55), αF-helix (Val174), β2-strand (Ile184), β3-strand (Val201), β4-strand (Ile221), β4–α3 loop (Leu232, Leu234), α3-helix (Ile253) and β5-strand (Ile264). These exchanging methyl groups could be grouped into three regions, the hinge region connecting the GTPase and helical domains (Ile55, Val174), switch 3 (Leu232, Leu234) and the β-sheets formed by β1, β3 and β4 (Ile184, Val201, Ile221, Ile253, Ile264). Although the methyl groups in direct contact with the bound GDP were not identified in the analyses, the existence of the conformational exchange processes is suggested in the adjacent β1–α1 loop (residues 41–45), which is called a phosphate-binding loop (P-loop) and directly interacts with the bound GDP through amide–phosphate interactions. In the $^1$H-$^{15}$N TROSY spectrum of Gα-GDP, the $^1$H line widths of the $^1$H-$^{15}$N amide resonances were significantly broadened in Ser44 and Gly45, with amide groups that form direct interactions with the β-phosphates of the bound GDP, as compared with those in other residues (Gly42 and Gly89), with amide groups that do not form any direct interactions with the bound GDP, suggesting that the conformational exchange processes accompanying the $^1$H chemical shift differences exist in these residues (Supplementary Fig. 7).

## The effect of $Mg^{2+}$ on Gα-GDP.
The results from the magnetic field-dependent MQ relaxation analyses revealed that Gα-GDP undergoes a conformational exchange process, and successfully identified the regions where significant structural rearrangements

occur. Together with the results of the MQ CPMG RD analyses, the timescales of the exchange processes are faster than those amenable to the CPMG RD approaches, and estimated to be on a microsecond order (the exchange rates of $10^3$–$10^4\,s^{-1}$). Recently, Goricanec et al.[33] demonstrated that Gα$_{i1}$Δ31·GDP exists in chemical exchange processes on a millisecond timescale (20–890$\,s^{-1}$), as revealed by the splitting of $^1$H-$^{15}$N amide resonances and the single quantum (SQ) CPMG RD analyses. These results are apparently inconsistent with our results, which show that Gα$_{i3}$·GDP gives a single set of $^1$H-$^{13}$C methyl resonances and exists in chemical exchange processes on a microsecond timescale ($10^3$–$10^4\,s^{-1}$) which are hardly detected by the CPMG RD analyses. We suppose that these differences are attributable to the different $Mg^{2+}$ concentrations in the NMR buffers.

Figure 3a,b show the overlays of the $^1$H-$^{15}$N TROSY and $^1$H-$^{13}$C HMQC spectra of a {u-$^2$H, $^{15}$N, Alaβ, Ileδ1-[$^{13}$CH$_3$]} Gα-GDP in the presence (red) and absence (black) of 20 mM $Mg^{2+}$. Upon the addition of $Mg^{2+}$, some NMR signals exhibited chemical shift changes. The apparent dissociation constant of $Mg^{2+}$ was calculated to be ∼3 mM, from the intensity ratio of the two signals (Fig. 3c). The methyl groups with significant chemical shift changes over 0.02 p.p.m. were Ile49 (α1-helix), Ile222 (β4-strand), Ile253 (α3-helix) and Ile264 (β5-strand). These residues were not clustered in the vicinity of the $Mg^{2+}$-binding site identified in the crystal structure (PDB ID: 1BOF) (ref. 34), suggesting that the $Mg^{2+}$ binding allosterically induces conformational changes in Gα-GDP (Fig. 3d). In the NMR analyses conducted by Goricanec et al.[33], the experiments were performed in the presence of 5 mM $Mg^{2+}$. Under these conditions, two signals would be observed in the $^1$H-$^{15}$N TROSY spectra. Indeed, the two Phe191 amide signals are very similar to the splitting of the $^1$H-$^{15}$N amide signal observed by Goricanec et al. (Fig. 7 in ref. 33).

Furthermore, we observed a chemical exchange process on a millisecond timescale in the presence of 5 mM $Mg^{2+}$. Figure 3e shows the results of the $^{13}$C SQ CPMG dispersion experiments of Ile49 and Ile222 in the presence and absence of 5 mM $Mg^{2+}$. The results clearly demonstrate that significant changes in the $R_{2,eff}$ rates are observed only in the presence of 5 mM $Mg^{2+}$. The fitted $\Delta\varpi_C$, $k_{ex}$ and $p_B$ values were calculated to be 0.44 p.p.m., $96\,s^{-1}$ and 0.12 for Ile49 and 0.26 p.p.m., $48\,s^{-1}$ and 0.22 for Ile222, respectively. Since the sizes of the $\Delta\varpi_C$ correlated with the chemical shift differences between the two signals observed in the presence of $Mg^{2+}$, we assume that the chemical exchange process observed in the presence of 5 mM $Mg^{2+}$ reflects the exchange process between the two conformations, induced by $Mg^{2+}$ binding. We also confirmed that the magnetic field-dependent changes in the $R_{MQ}$ and $\Delta R_{MQ}$ rates were still observed in the presence of $Mg^{2+}$. These results support our proposal that the millisecond exchange processes observed in the presence of $Mg^{2+}$ are different exchange processes from those observed in the magnetic field-dependent MQ relaxation analyses, and these distinct exchange processes are probably related to the different functions of Gα.

Thus, we conclude that the differences in the results of the CPMG RD experiments are mainly due to different $Mg^{2+}$ concentrations, and that the existence of the chemical exchange on a millisecond timescale is consistent with the results of the CPMG RD experiments by Goricanec et al.[33], which were performed in the presence of 5 mM $Mg^{2+}$. However, it should be noted that we did not detect significant changes in the $R_{2,eff}$ rates in the methyl groups from the helical domain (Leu159) and the C-terminal region (Val342) in the presence of 5 mM $Mg^{2+}$, although significant changes in the $R_{2,eff}$ rates were observed for these methyl groups in the SQ CPMG RD analyses by Goricanec.

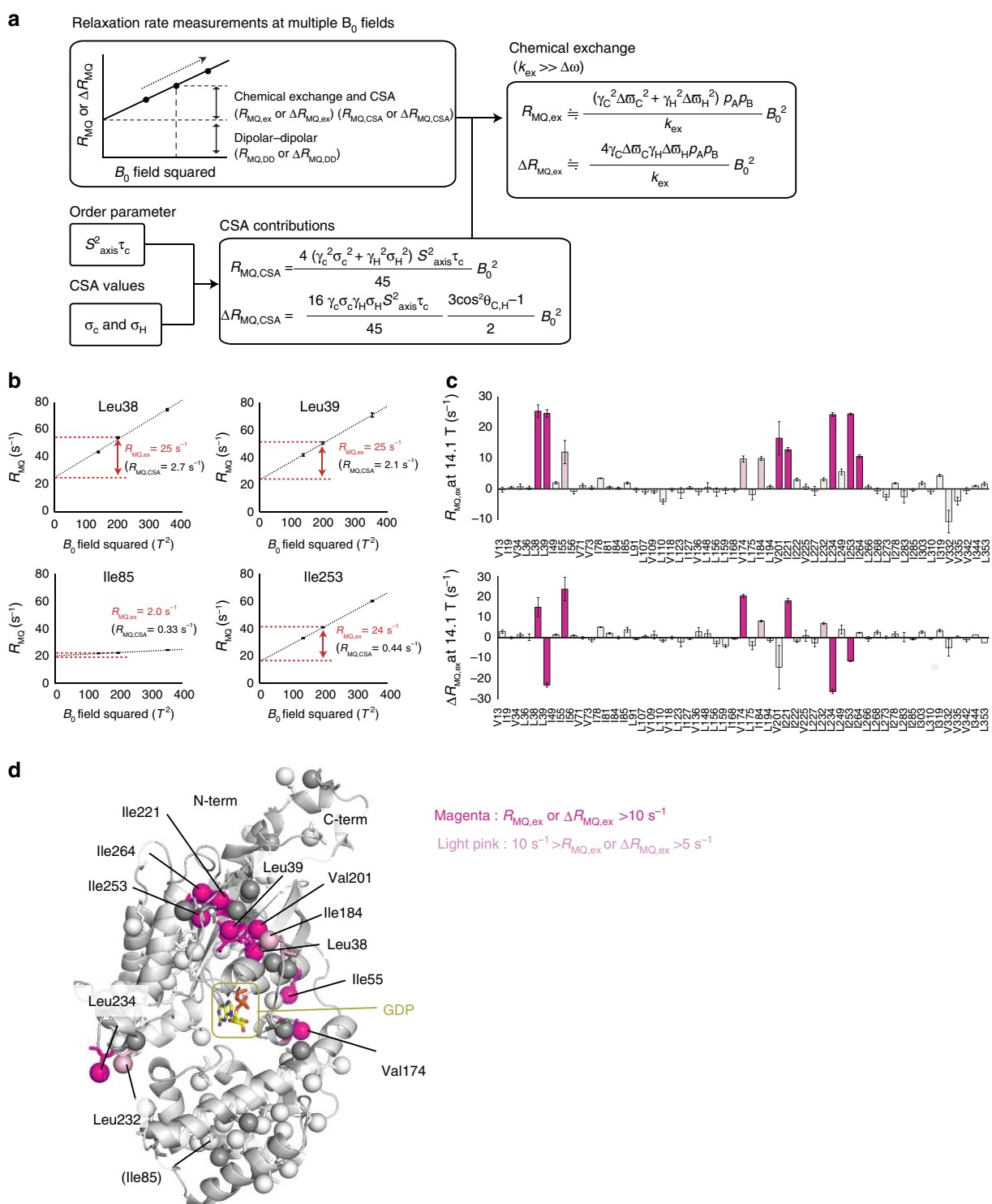

**Figure 2 | Magnetic field-dependent MQ relaxation analyses of the wild-type Gα-GDP.** (**a**) Schematic representation of the magnetic field-dependent MQ relaxation analyses. (**b**) Plots of the $R_{MQ}$ rates of Leu38, Leu39, Ile85 and Ile253 against the square of the static magnetic fields. The obtained $R_{MQ,ex}$ rates at 14.1 Tesla (600 MHz ¹H frequency) are shown. The error bars represent s.d. of fitting errors, estimated from Monte Carlo simulations using uncertainties in peak intensities. (**c**) Plots of the $R_{MQ,ex}$ (top) and $\Delta R_{MQ,ex}$ (bottom) rates at 14.1 Tesla. The error bars represent s.d. of fitting errors, estimated from Monte Carlo simulations using uncertainties in peak intensities. The methyl groups with $R_{MQ,ex}$ or $\Delta R_{MQ,ex}$ rates larger than 10 s⁻¹ are coloured magenta, and those within the 5–10 s⁻¹ range are coloured pink. The small negative values of the $R_{MQ,ex}$ rates are due to the overestimations to the CSA contributions to the MQ relaxation rates, originating from systematic errors in the measurements. We suppose that the relatively large negative value of the $R_{MQ,ex}$ rate of Val332 is due to its low signal-to-noise ratio and its very large dipolar contribution to the MQ relaxation rate (44 s⁻¹), which hampers the reliable measurement of the ¹H-¹³C cross-correlated relaxation rate (1.9 s⁻¹ on average). (**d**) Mapping of the methyl groups with significant $R_{MQ,ex}$ or $\Delta R_{MQ,ex}$ rates on the structure of Gα-GDP (PDB ID: 1GDD)[17]. The methyl groups with significant $R_{MQ,ex}$ or $\Delta R_{MQ,ex}$ rates are coloured in the same manner as in **c**. Methyl groups with no data are coloured grey.

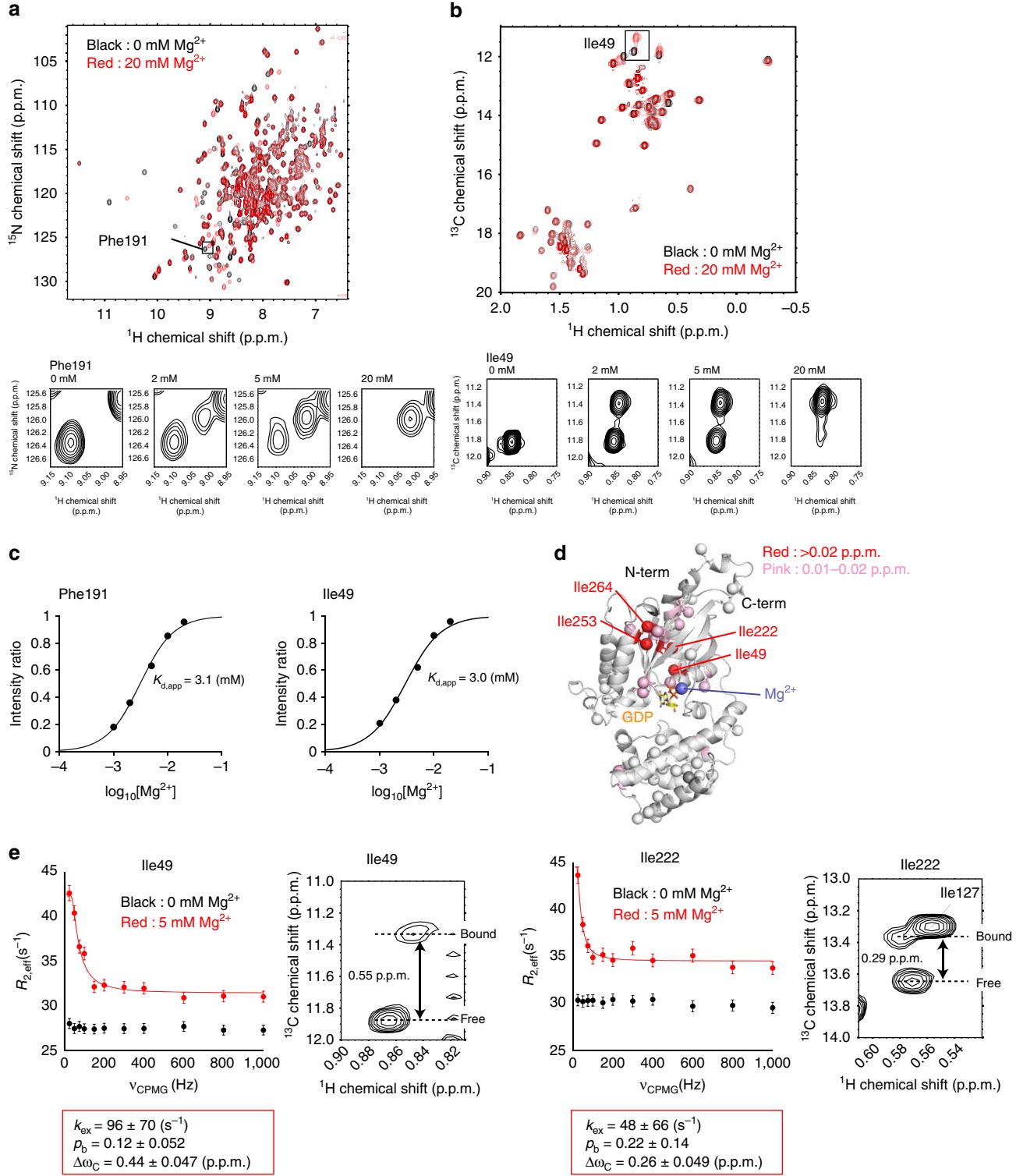

**Figure 3 | Effects of Mg²⁺ binding on Gα-GDP.** (**a**) Overlay of the ¹H-¹⁵N TROSY spectra of {u-²H, ¹⁵N, Alaβ, Ileδ1-[¹³CH₃]} Gα-GDP in the presence (red) and absence (black) of 20 mM Mg²⁺. The Mg²⁺ concentration-dependence of the Phe191 signal is shown below. (**b**) Overlay of the ¹H-¹³C HMQC spectra of {u-²H, ¹⁵N, Alaβ, Ileδ1-[¹³CH₃]} Gα-GDP in the presence (red) and absence (black) of 20 mM Mg²⁺. The Mg²⁺ concentration-dependence of the Ile49 signal is shown below. (**c**) The intensity ratios of the two signals observed for Phe191 and Ile49 were plotted against the Mg²⁺ concentration. The apparent dissociation constants calculated from the Phe191 and Ile49 signals were 3.1 and 3.0 mM, respectively. (**d**) The Ala and Ile methyl groups with significant chemical shift differences are mapped on the crystal structure of Gα-GDP bound to Mg²⁺ (PDB ID: 1BOF)[34]. The methyl groups with chemical shift differences larger than 0.02 p.p.m. are coloured red, and those within the 0.01 − 0.02 p.p.m. range are coloured pink. (**e**) ¹³C SQ CPMG dispersion experiments of Ile49 and Ile222. The results in the presence of 5 mM Mg²⁺ are coloured red, and those in the absence of Mg²⁺ are coloured black. The error bars represent the experimental errors calculated using the equation (16). The fitted parameters are summarized below. The measurements were performed at 20 °C with a Bruker Avance 600 spectrometer. The ¹H-¹³C HMQC spectra of the corresponding residues are also shown.

We suppose these variations are originated from the differences in the construct ($G\alpha_{i1}\Delta 31$ and $G\alpha_{i3}$), the buffer conditions and the temperature, and further experimental studies are required to clarify these points.

Since the intracellular free $Mg^{2+}$ concentration is reportedly within the range of 0.2–1.2 mM (refs 35,36), the majority of $G\alpha$-GDP ($>70\%$) is estimated to be in the $Mg^{2+}$-unbound state under physiological conditions. Therefore, we emphasize that our results have been obtained under physiologically relevant conditions. In addition, we confirm that the absence of $Mg^{2+}$ does not preclude the analysis of the GDP dissociation process, because $Mg^{2+}$ reportedly does not affect the rate of GDP dissociation from $G\alpha$ (ref. 37).

**MQ relaxation analyses of the D150N mutant.** We applied the magnetic field-dependent MQ relaxation analyses to the D150N mutant, which exhibited the faster dissociation of GDP, and compared the results with those from the wild type. Figure 4a shows the plots of the $R_{MQ,ex}$ rates of the D150N mutant at 14.1 Tesla (600 MHz $^1$H frequency). While significantly large $R_{MQ,ex}$ rates were observed in the same methyl groups as the wild type, the $R_{MQ,ex}$ rates in Leu38, Leu39 and Val201 were larger than those of the wild type. These methyl groups are clustered on the $\beta 1$-strand (Leu38, Leu39) and the adjacent $\beta 3$-strand (Val201; Fig. 4b), indicating that the conformational exchange processes in the $\beta 1$ strand are facilitated in the D150N mutant. In the crystal structure of $G\alpha$-GDP (ref. 17), the side-chain of Asp150 forms a hydrogen bond network with the Lys270 ($\beta 5$-strand) side-chain and the Glu43 (P-loop) main-chain carbonyl group. Therefore, the substitution of Asp to Asn could cause the disruption of the network, leading to the altered dynamics in the P-loop and adjacent $\beta 1$-strand.

To further characterize the exchange process, we analysed the peak positions of Leu38, Leu39 and Val201. In the D150N mutant, the peak positions were slightly different from those in the wild type (Fig. 4c). If the observed differences in the peak positions reflect the shift in the conformational equilibrium in the D150N mutant, then the corresponding $^1$H-$^{13}$C MQ chemical shift differences, that is ($\gamma_C^2 \Delta \varpi_C^2 + \gamma_H^2 \Delta \varpi_H^2$; equation (1)) and $4\gamma_C \Delta \varpi_C \gamma_H \Delta \varpi_H$; equation (2)), should be correlated to the $R_{MQ,ex}$ and $\Delta R_{MQ,ex}$ rates, respectively. Indeed, the MQ chemical shift differences calculated from the differences in the peak positions were highly correlated to the $R_{MQ,ex}$ and $\Delta R_{MQ,ex}$ rates ($R = 0.88$), strongly suggesting that the increase in the $R_{MQ,ex}$ rates reflects the shift in the conformational equilibrium, and that the minor state population is increased in the D150N mutant (Fig. 4d).

**The effect of the binding of the GoLoco14 peptide.** To further investigate the relationship between the GDP dissociation and the conformational exchange processes in $G\alpha$-GDP, we subsequently applied the method to $G\alpha$-GDP bound to the GoLoco14 peptide, the $G\alpha$-binding motif of the RGS14 protein[38,39]. The GoLoco14 peptide selectively binds to the $G\alpha$-GDP state in the signalling cascade, and inhibits the GDP dissociation. Therefore, we investigated the conformational exchange processes in $G\alpha$-GDP bound to the GoLoco14 peptide, where the GDP dissociation is suppressed.

We prepared a peptide containing the GoLoco14 sequence (GoLoco14), and confirmed that the $^3$H-GDP dissociation was inhibited in its presence (Fig. 5a). We measured the IC50 value of GoLoco14, from the concentration dependency of the inhibition of the $^3$H-GDP dissociation. The IC50 of GoLoco14 was calculated to be $620 \pm 150$ nM, consistent with the previous report of IC50 values in the sub-micromolar range[38] (Fig. 5b). The binding of GoLoco14 was also confirmed from the chemical

shift changes observed in the $^1$H-$^{13}$C HMQC spectra upon the addition of GoLoco14 (Supplementary Fig. 8).

We applied the magnetic field-dependent MQ relaxation analyses to GoLoco14-bound $G\alpha$-GDP (GoLoco14-$G\alpha$), and compared the results with those obtained in the absence of GoLoco14. Figure 5c shows the plots of the $R_{MQ,ex}$ rates of GoLoco14-$G\alpha$ at 14.1 Tesla (600 MHz $^1$H frequency), in comparison with the results obtained in the absence of GoLoco14. Strikingly, the $R_{MQ,ex}$ rates in GoLoco14-$G\alpha$ were smaller than those in the absence of GoLoco14, for the majority of the methyl groups. The methyl groups that showed the reduced $R_{MQ,ex}$ rates were Leu38, Leu39, Ile55, Val174, Ile221 and Leu234. These methyl groups are located on the $\beta 1$-strand (Leu38, Leu39), $\alpha 1$-helix (Ile55), $\alpha F$-helix (Val174) and $\beta 4$–$\alpha 3$ loop (Leu234), indicating that the conformational exchange processes in these regions are suppressed by the binding of GoLoco14 (Fig. 5d). Notably, the conformational exchange processes in the $\beta 1$ strand (Leu38, Leu39) were suppressed by the binding of GoLoco14, in contrast to the results obtained with the D150N mutant, in which the conformational exchange processes in the $\beta 1$-strand were facilitated. These results strongly suggest that the conformational exchange processes in the $\beta 1$-strand are closely related to the GDP dissociation.

**Discussion**

Here, we developed magnetic field-dependent MQ relaxation analyses for characterizing the chemical exchange processes on a microsecond timescale in high molecular weight proteins. The results from $G\alpha$-GDP revealed that the method is particularly beneficial for characterizing the chemical exchange processes on a microsecond timescale, which are difficult to detect by the conventional CPMG RD experiments. Moreover, the method utilizes the measurements of the $^1$H-$^{13}$C MQ relaxation rates and benefits from the methyl-TROSY principle, enabling the analyses of conformational equilibrium processes in high molecular weight proteins. Although the extraction of the exchange parameters ($k_{ex}, p_B$ and $\Delta \varpi$) is usually difficult to achieve solely from the observed $R_{MQ,ex}$ and $\Delta R_{MQ,ex}$ rates, the high sensitivities to the existence of chemical exchange processes[40,41] and the linearity in the MQ chemical shift differences (equations (1) and (2)) would greatly facilitate the characterization of the conformational exchange phenomena. It is becoming increasingly apparent that conformational exchange processes on a microsecond timescale are closely related to various kinds of protein functions, such as molecular recognition[27,42,43], ligand binding[44], signal transduction[45] and receptor activation[46]. So far, the analyses of the exchange processes on a microsecond timescale have been restricted to small- and medium-sized proteins, due to the limited sensitivities of the existing methods. Applications of the magnetic field-dependent MQ relaxation analyses will accelerate the elucidations of the functional dynamics of biologically significant proteins with high molecular weights, such as membrane proteins and functional protein complexes.

We applied the magnetic field-dependent MQ relaxation analyses to $G\alpha$-GDP, and identified the regions that exist in a conformational exchange process. Our analyses revealed that the conformational exchange processes observed in the $\beta 1$-strand are facilitated in the D150N mutant, which exhibits the accelerated GDP dissociation, while the exchange processes are suppressed by the binding of GoLoco14, which inhibits the GDP dissociation. These results suggest that the conformational exchange processes in the $\beta 1$-strand play a fundamental role in the GDP dissociation.

As shown from the comparisons between the $R_{MQ,ex}$ and $\Delta R_{MQ,ex}$ rates and the differences in the MQ chemical shifts,

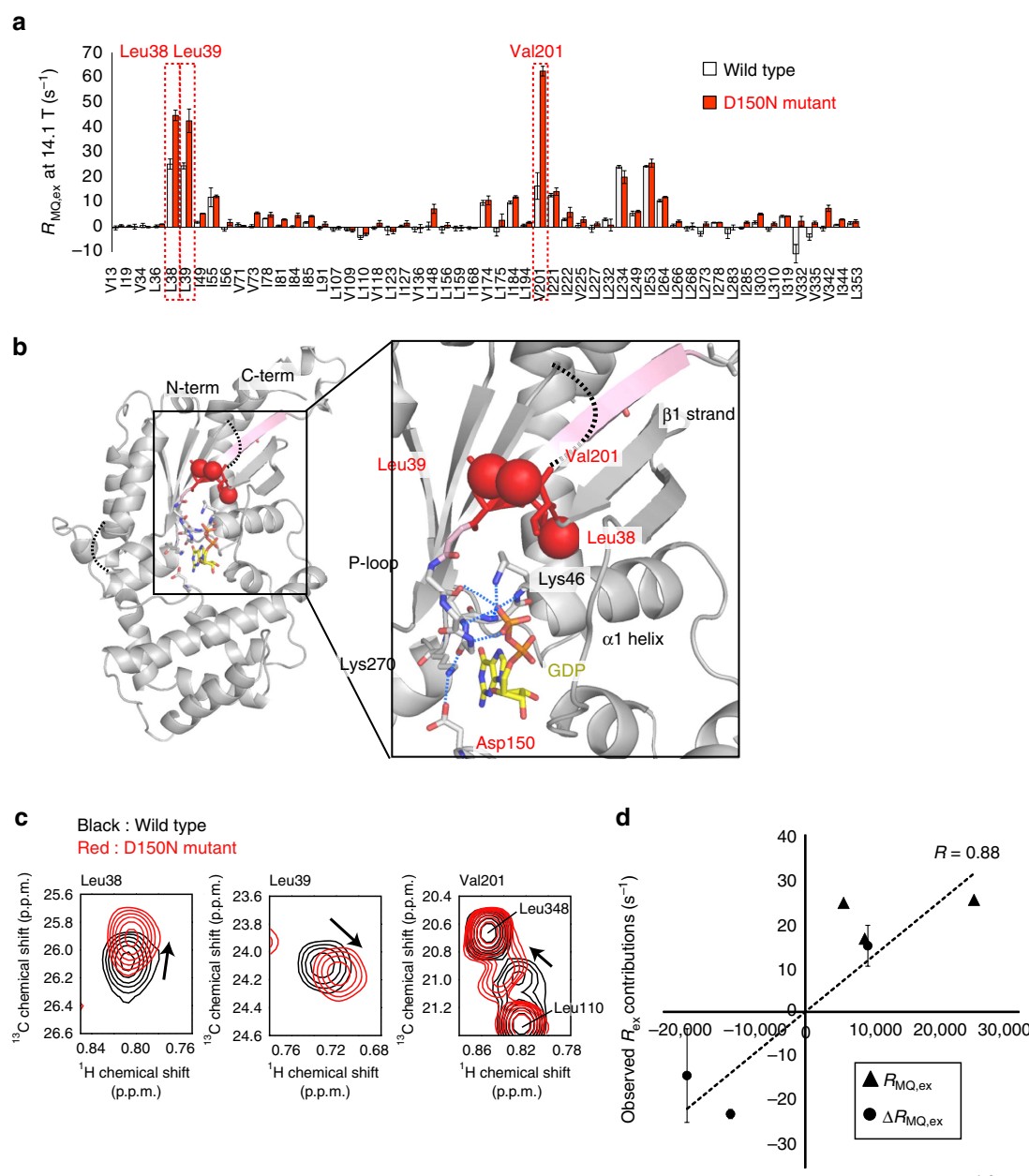

**Figure 4 | Magnetic field-dependent MQ relaxation analyses of the D150N mutant. (a)** Plots of the $R_{MQ,ex}$ rates of the wild type (light grey) and those of the D150N mutant (red) at 14.1 Tesla (600 MHz $^1$H frequency). The error bars represent s.d. of fitting errors, estimated from Monte Carlo simulations using uncertainties in peak intensities. The methyl groups with increased $R_{MQ,ex}$ rates in the D150N mutant are highlighted. **(b)** Expanded view of the Leu38, Leu39 and Val201 methyl groups in the crystal structure of Gα-GDP (PDB ID: 1GDD)[17]. The methyl carbons are shown as red spheres. Lys46, Lys270 and Asp150 are shown as sticks. **(c)** The overlay of the $^1$H-$^{13}$C HMQC spectra of Leu38, Leu39 and Val201. The spectra of the wild type are coloured black, and those of the D150N mutant are coloured red. **(d)** Linear correlation plot of the $R_{MQ,ex}$ (triangles) and $\Delta R_{MQ,ex}$ rates (circles) against the MQ chemical shift differences between the wild type and the D150N mutant for Leu38, Leu39 and Val201. The error bars represent s.d. of fitting errors, estimated from Monte Carlo simulations using uncertainties in peak intensities.

the fraction of the minor state in the equilibrium seems to be increased in the D150N mutant, as compared with the wild-type. Taken together, we propose the regulatory mechanism of the GDP dissociation driven by the conformational exchange, as summarized in Fig. 6. The β1-strand region of Gα-GDP exists in a conformational equilibrium, and the minor conformation in the equilibrium has significantly reduced affinity to GDP. The ground-state conformation is assumed to be stabilized by the hydrogen bond network formed by Asp150, Lys270 and Glu43, which anchor the β1-strand and the P-loop to the bound GDP as

observed in the crystal structure (Figs 4b and 6b). In the D150N mutant, the ground-state conformation is destabilized by weakening of the hydrogen bond network, due to the loss of the negative charge on the side chain, and the conformational equilibrium shifts towards the minor state, leading to the accelerated dissociation of GDP (Fig. 6c). However, the binding of GoLoco14 suppresses the conformational exchange processes in the β1-strand and stabilizes the ground-state conformation, which tightly binds to GDP, leading to the inhibition of the GDP dissociation (Fig. 6a). The analyses of the $^1$H line widths of the

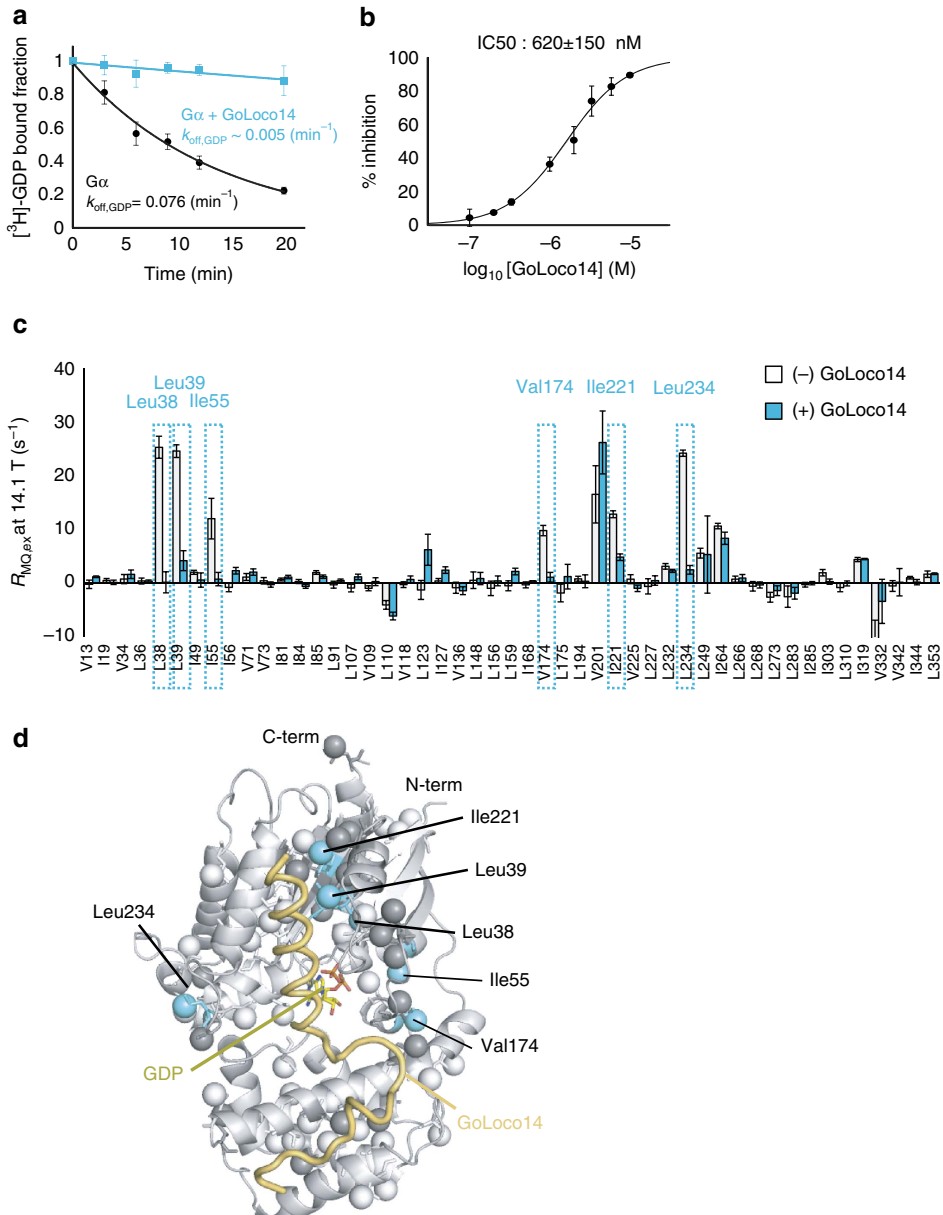

**Figure 5 | Effects of the binding of GoLoco14. (a)** $^3$H-GDP dissociation assays of Gα-GDP in the presence (cyan) and absence (black) of 10 µM GoLoco14. The experiments were performed at 30 °C. Each point reflects mean ± s.e.m. of three independent experiments. **(b)** Plot of the percentage inhibition of the $^3$H-GDP dissociation against the concentration of GoLoco14. The percentage inhibition was calculated from the amount of bound $^3$H-GDP after a 10 min incubation in the presence of various concentrations of GoLoco14 (100 nM to 10 µM). The calculated IC50 value was 620 ± 150 nM. Each point reflects mean ± s.e. of three independent experiments. **(c)** Plots of the $R_{MQ,ex}$ rates in the presence (cyan) and absence (light grey) of GoLoco14 at 14.1 Tesla (600 MHz $^1$H frequency). The error bars represent s.d. of fitting errors, estimated from Monte Carlo simulations using uncertainties in peak intensities. The methyl groups with reduced $R_{MQ,ex}$ rates in the presence of GoLoco14 are highlighted. We suppose that the increase in the $R_{MQ,ex}$ rates of Val201 is mainly due to the change in the $\Delta\varpi$ value, induced by the binding of GoLoco14 ($\sim$0.4 p.p.m., Supplementary Fig. 8b). **(d)** Mapping of the methyl groups with reduced $R_{MQ,ex}$ rates in the presence of GoLoco14 on the structure of GoLoco14-Gα·GDP (PDB ID: 1KJY)[39]. The methyl groups with reduced $R_{MQ,ex}$ rates in the presence of GoLoco14 are coloured cyan. Methyl groups with no data are coloured grey.

amide resonances support that the conformational equilibrium processes exist in the P-loop region, which directly interacts with the bound GDP through amide–phosphate interactions. Therefore, the reduced affinity to GDP in the minor state conformation seems to be mainly attributed to the altered interaction mode of the P-loop with the GDP phosphates. Our proposed scheme indicates that the dissociation of GDP is driven by the conformational dynamics of Gα, and may be generally applied to the functional mechanisms of other Gα mutants and the ligands that modulate Gα functions.

Rearrangements in the hydrogen bond network formed by Asp150 have not been observed in the crystal structures with the reduced affinity to GDP[14,15]. Therefore, it has been difficult to elucidate the structural mechanism of the accelerated GDP dissociation in the D150N mutant by simply comparing the crystal structures. The crystal structures represent the ground-state structures tightly bound to GDP, and provide the limited insights into the excited state structures, which is only transiently formed and play critical roles in the GDP dissociation. In contrast, the NMR results reveal the existence of conformational

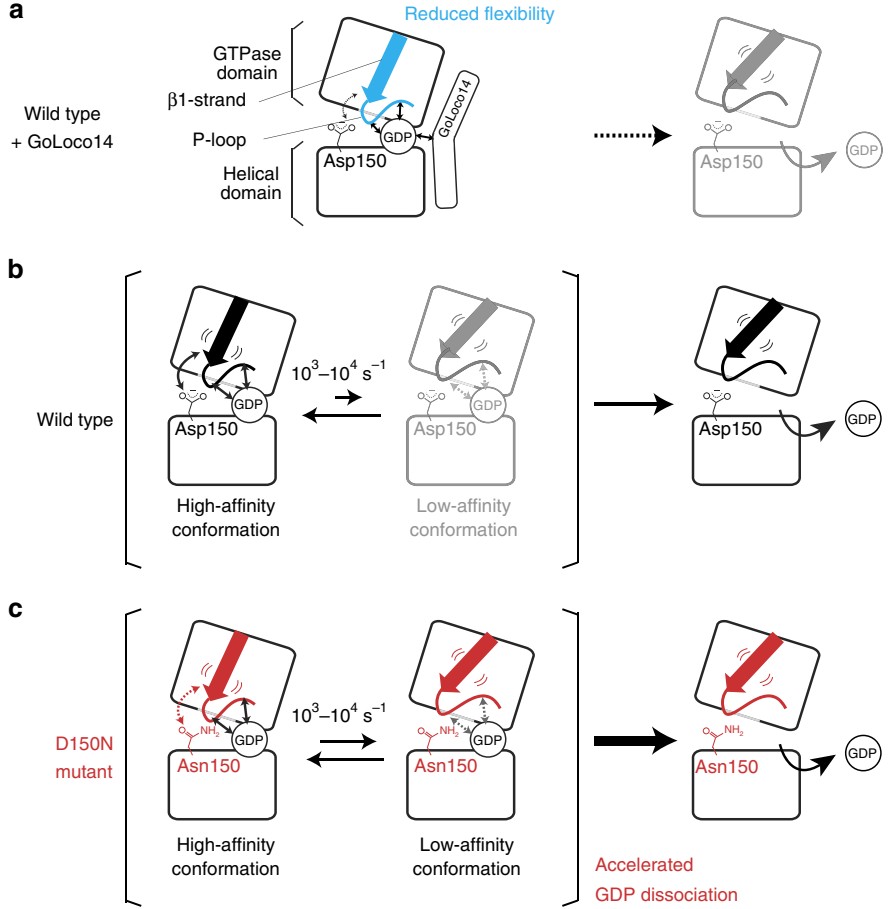

**Figure 6 | Schematic representation of the conformational equilibrium in Gα-GDP and the regulation of the GDP dissociation. (a)** The binding of GoLoco14 suppresses the conformational exchange processes and stabilizes the ground-state conformation that tightly binds to GDP, leading to the inhibition of the GDP dissociation. **(b)** The β1-strand and P-loop regions exist in an equilibrium between conformations with high and low affinity to GDP. The low-affinity conformation promotes the GDP dissociation. **(c)** The conformational equilibrium shifts towards the low-affinity conformation in the D150N mutant, causing the accelerated GDP dissociation.

equilibrium processes, and provide structural insights into the excited state structure from the measurements of the $R_{MQ,ex}$ and $\Delta R_{MQ,ex}$ contributions. In this point of view, our NMR study advances the mechanistic understanding in the functions of the G proteins, and precisely complements the crystallographic studies so far.

Our NMR results also indicate that the conformational exchange processes exist in the switch regions. Studies of the small GTPase Ras have revealed that the GTP-bound Ras is exchanging between different conformational states, called state 1 and state 2, which show structural differences in switch 1 and switch 2, and that the transitions between these two states are closely related to the guanine nucleotide association/dissociation kinetics[47–49]. Given the high structural homology between the Gα-GTPase domain and Ras, the conformational exchange processes observed in Gα might correpond to the state 1 to state 2 transition in Ras.

Although our study has focused on the GDP dissociation from Gα, in the absence of the catalytic activity of an activated GPCR, the modulation of the conformational dynamics in the β1 strand is also suggested to occur in the GPCR-catalysed GDP dissociation. Peptide amide hydrogen-deuterium exchange mass spectroscopy analyses, using activated β2-adrenergic receptor (β₂AR) and Gαs proteins, identified the β1-strand as the region where structural rearrangements occur upon GPCR activation[50]. Since the structural differences observed in the β1-strand were

very small between the crystal structures of Gα-GDP and the β₂AR·Gα complex[51], the role of the structural rearrangements in the β1-strand has remained unclear. Considering our NMR results, the activated GPCR may promote the spontaneous GDP release, by modulating the conformational equilibrium within Gα and stabilizing the minor state conformation with the reduced GDP affinity.

In summary, we established the magnetic field-dependent NMR relaxation analyses, and revealed that Gα exchanges between a ground-state conformation, which tightly binds to GDP, and an excited conformation with reduced affinity to GDP. The oncogenic mutation and the binding of GoLoco14 regulate the dissociation of GDP by modulating the conformational equilibrium processes within Gα. We also demonstrated that the structural rearrangements in the β1-strand play critical roles in regulating the binding affinity for GDP. The mechanism described here provides the structural insight into the activation mechanism of G proteins, and demonstrates the importance of the conformational dynamics of Gα in regulating G protein signalling.

## Methods
**Protein expression and purification.** The human Gα<sub>i3</sub> (residues 1–354) protein, including an N-terminal His10-tag and an HRV-3C protease recognition site, was expressed in *Escherichia coli* BL21 (DE3) cells (Agilent Technologies). For the selective $^{13}CH_3$-labelling of methyl groups, *E. coli* cells were grown in deuterated

M9 media and $50\,mg\,l^{-1}$ of [methyl-$^{13}$C, 3-$^2$H$_2$]-$\alpha$-ketobutyric acid (for Ile$\delta$1) (CIL) and $300\,mg\,l^{-1}$ of [2-methyl $^{13}$C, 4-$^2$H$_3$]-acetolactate (for Leu$\delta$2 and Val$\gamma$2) (NMR-Bio) were both added 1 h before the induction[52,53]. The G$\alpha$ protein was purified by affinity chromatography using a His 10-tag (refs 54,55). Briefly, the protein was purified to homogeneity by chromatography on HIS-Select resin (Sigma-Aldrich). After cleavage of the His10-tag with HRV-3C protease (Novagen), the cleaved His-tags and the protease were removed with HIS-Select resin. The NMR sample consisted of 0.20 mM G$\alpha$-GDP, 20 mM HEPES-NaOH (pH 7.0), 0.5 mM GDP and 5 mM DTT in D$_2$O. The D150N mutant was constructed with a QuikChange Site-directed Mutagenesis Kit (Agilent Technologies).

The GoLoco14 peptide (residues 496–531 of rat RGS14), including an N-terminal glutathione S-transferase (GST)-tag, was expressed in *E. coli* C41 (DE3) cells (Lucigen). The peptide was purified with Glutathione sepharose (GE). After the cleavage of the GST-tag with HRV-3C protease (Novagen), the peptide was further purified using a Superdex peptide column (Pharmacia). The peptide contains five additional N-terminal amino acids (Gly-Pro-Leu-Gly-Ser) derived from the HRV-3C protease recognition sequence. The mass of the peptide was confirmed by MALDI-TOF mass spectrometry, using an Axima TOF 2 mass spectrometer (Shimadzu Biotech).

**$^3$H-GDP dissociation rate measurements.** The dissociation rate of GDP was measured by monitoring the reduction in the amount of radio-labelled GDP bound to G$\alpha$ (ref 56). The purified G$\alpha$-GDP protein or its mutant (200 nM) was incubated at 20 °C for 3 h, in a buffer containing 20 mM HEPES-NaOH (pH 7.0), 200 mM NaCl, 5 mM DTT and 1 $\mu$M [8, 5′-$^3$H]-GDP (Perkin Elmer). The exchange reaction was initiated by adding an excess amount of unlabelled GDP (100 $\mu$M final concentration). At the indicated time points, aliquots were collected and passed through a NAP-5 size-exclusion column (GE) to remove the free GDP. The amount of bound $^3$H-GDP was determined by liquid scintillation counting.

**NMR analyses.** All experiments were performed on Bruker Avance 500, 600 or 800 spectrometers equipped with cryogenic probes. All spectra were processed by the Bruker TopSpin 2.1 or 3.1 software, and the data were analysed by Sparky (T.D. Goddard and D.G. Kneller, Sparky 3, University of California, San Francisco, CA, USA).

**Establishment of MQ relaxation analyses.** To evaluate the chemical exchange contributions to the MQ relaxation rates, we exploited the fact that the exchange contributions increase linearly with the square of the static magnetic field strengths in a fast exchange regime. The magnetic field-dependency has been utilized for characterizing exchange processes, by observing the backbone $^1$H-$^{15}$N resonances[28–30]. In the side-chain methyl groups, the $^1$H-$^{13}$C MQ relaxation rates $R_{MQ}$ ($=0.5*(R_{ZQ}+R_{DQ})$) and the differential MQ relaxation rates $\Delta R_{MQ}$ ($=R_{DQ}-R_{ZQ}$) can be expressed as the sum of the dipolar contributions (with the subscript DD), the CSA contributions (with the subscript CSA), and the exchange contributions (with the subscript ex) (equations (3) and (4))[57–60].

$$R_{MQ} = R_{MQ,DD} + R_{MQ,CSA} + R_{MQ,ex} \qquad (3)$$

$$\Delta R_{MQ} = \Delta R_{MQ,DD} + \Delta R_{MQ,CSA} + \Delta R_{MQ,ex} \qquad (4)$$

Here, we assume a slow tumbling limit ($\omega^2\tau_c^2 \gg 1$, where $\omega$ represents Larmor frequency and $\tau_c$ represents rotational correlation time) and a fast exchange regime (exchange rates, $k_{ex} \gg$ chemical shift differences, $\Delta\omega$). In the case of proteins with molecular weights larger than 20 K (corresponding to the correlation time of $\sim 12$ ns), the value of $\omega^2\tau_c^2$ is estimated to be larger than $10^2$. In addition, the MQ chemical shift differences in $^{13}$CH$_3$ methyl groups are usually smaller than 500 Hz at 14.1 Tesla, as judged from the case of T4 lysozyme mutant, which is below the exchange rates of the microsecond exchange processes with the order of $10^3$–$10^5$ s$^{-1}$. Therefore, the assumptions of a slow tumbling limit and a fast exchange regime usually hold for the microsecond exchange processes in high molecular weight proteins. Under these assumptions, the CSA and exchange contributions are both proportional to the square of the static magnetic field strength, $B_0$.

The CSA contributions, $R_{MQ,CSA}$ and $\Delta R_{MQ,CSA}$, can be expressed by equations (5) and (6), using the gyromagnetic ratios, $\gamma_C$ and $\gamma_H$, the CSA values, $\Delta\sigma c$ and $\Delta\sigma_H$, the methyl threefold axis order parameter, $S^2_{axis}$, the rotational correlation time, $\tau_C$, and the angle between the principal axis of the CSA tensor, $\theta_{C,H}$[57,58].

$$R_{MQ,CSA} = \frac{4(\gamma_C^2\Delta\sigma_C^2 + \gamma_H^2\Delta\sigma_H^2)S^2_{axis}\tau_C}{45} \cdot B_0^2 \qquad (5)$$

$$\Delta R_{MQ,CSA} = \frac{16\gamma_C\Delta\sigma_C\gamma_H\Delta\sigma_H S^2_{axis}\tau_C}{45} \cdot \frac{3\cos^2\theta_{C,H}-1}{2} \cdot B_0^2 \qquad (6)$$

Assuming a simple two-state (states A and B) chemical exchange process, the exchange contributions, $R_{MQ,ex}$ and $\Delta R_{MQ,ex}$, can be expressed by equations (1) and (2) in the main text[31,32]. The $R_{MQ,ex}$ rates are proportional to the sum of $\gamma_C^2\Delta\varpi_C^2$ and $\gamma_H^2\Delta\varpi_H^2$, while the $\Delta R_{MQ,ex}$ rates are proportional to the product of $\gamma_C\Delta\varpi_C$ and

$\gamma_H\Delta\varpi_H$. Therefore, the analysis of both the $R_{MQ,ex}$ and $\Delta R_{MQ,ex}$ rates would provide detailed information about the exchange processes, such as the relative signs of $\Delta\varpi_C$ and $\Delta\varpi_H$.

On the basis of the equations (1)–(6), the chemical exchange contributions can be extracted using (i) the $R_{MQ}$ and $\Delta R_{MQ}$ rates measured at different static magnetic fields, (ii) the CSA values, $\Delta\sigma c$ and $\Delta\sigma_H$ and (iii) the products of the order parameter and the rotational correlation time, $S^2_{axis}\tau_C$, as summarized in Fig. 2a.

The $R_{MQ}$ and $\Delta R_{MQ}$ rates are measured by Hahn-echo type relaxation measurements, established by Gill and Palmer[61]. During the Hahn-echo relaxation period $T$ ($=n/2J_{CH}$), double quantum (DQ) or zero quantum (ZQ) coherences were selected, and the relaxation rates were extracted by fitting the peak intensities to an exponential decay function (equation (7)). $I(T)$ represents the peak intensities with the relaxation period $T$, and $R$ represents the relaxation rates ($R_{DQ}$ or $R_{ZQ}$).

$$I(T) = I(0) \times \exp(-RT) \qquad (7)$$

The $R_{MQ}$ and $\Delta R_{MQ}$ rates were calculated using equation (8).

$$R_{MQ} = 0.5 \times (R_{DQ} + R_{ZQ})$$
$$\Delta R_{MQ} = R_{DQ} - R_{ZQ} \qquad (8)$$

In current applications, the $R_{MQ}$ and $\Delta R_{MQ}$ rates were obtained at three or four static magnetic fields, ranging from 9.4 Tesla (400 MHz $^1$H frequency) to 18.8 Tesla (800 MHz $^1$H frequency).

The CSA values, $\Delta\sigma c$ and $\Delta\sigma_H$, were measured from the $^1$H-$^{13}$C dipolar-CSA cross-correlated relaxation rates[62,63].

For measuring the $^{13}$C CSA values, $^1$H-coupled constant-time heteronuclear SQ coherence spectra were recorded with varied constant-time periods $T$, then the relaxation rates of the inner two multiplet components during $T$ were extracted by fitting the peak intensities to an exponential decay function. The $^{13}$C cross-correlated relaxation rates, $\eta_C$, were calculated using equation (9), where $R_{C,uf}$ and $R_{C,df}$ represent the relaxation rates of the up-field and down-field multiplet components, respectively. The $\eta_C$ can be expressed using the methyl threefold axis order parameter, $S^2_{axis}$, the rotational correlation time, $\tau_C$, the gyromagnetic ratios, $\gamma_C$ and $\gamma_H$, the static magnetic field strength, $B_0$, the CSA values, $\Delta\sigma'_C$, the angle between the HC bond and the methyl threefold axis, $\beta$, the length of the methyl $^1$H-$^{13}$C bond, $r_{HC}$ and $P_2(\cos x) = (3\cos^2 x-1)/2$. We used the $P_2(\cos\beta)/r^3_{HC}$ value of $-0.228$ Å$^{-3}$. The $\Delta\sigma'_c$ value of each methyl group was calculated using the $S^2_{axis}\tau_c$ (see the next section) and the $\eta_C$ rate.

$$\begin{aligned}\eta_C &= 0.5 \times (R_{C,uf} - R_{C,df}) \\ &= -\frac{4}{15}\left(\frac{\mu_0}{4\pi}\right)S^2_{axis}\tau_C\gamma_H\gamma_c B_0\hbar\Delta\sigma'_C\frac{P_2(\cos\beta)}{r^3_{HC}}\end{aligned} \qquad (9)$$

For measuring the $^1$H CSA values, $^{13}$C-coupled HMQC spectra were recorded with varied delays $T$, and then the relaxation rates of two multiplet components during $T$ were extracted by fitting the peak intensities to an exponential decay function. The $^1$H cross-correlated relaxation rates, $\eta_H$, were calculated using equation (10), where $R_{H,uf}$ and $R_{H,df}$ represent the relaxation rates of the up-field and down-field multiplet components, respectively. The CSA value, $\Delta\sigma'_H$, of each methyl group was calculated using the $S^2_{axis}\tau_c$ and the $\eta_H$ rate.

$$\begin{aligned}\eta_H &= 0.5 \times (R_{H,uf} - R_{H,df}) \\ &= -\frac{4}{15}\left(\frac{\mu_0}{4\pi}\right)S^2_{axis}\tau_C\gamma_H^2 B_0\hbar\Delta\sigma'_H\frac{P_2(\cos\beta)}{r^3_{HC}}\end{aligned} \qquad (10)$$

When we used the CSA values, $\Delta\sigma'c$ and $\Delta\sigma'_H$, to estimate the $R_{MQ,CSA}$ and $\Delta R_{MQ,CSA}$ rates, the calculated rates tended to be slightly smaller than the experimentally observed rates, probably due to the methyl threefold rotational averaging of the CSA values. Therefore, we used the scaled CSA values, $\Delta\sigma c$ and $\Delta\sigma_H$, in equation (11) when we calculated the $R_{MQ,CSA}$ and $\Delta R_{MQ,CSA}$ rates. We assume that the principal axes of the $^{13}$C and $^1$H CSA tensors are parallel, $(3\cos^2\theta_{C,H}-1)/2=1$ (equation (6)). These scaling factors were experimentally determined.

$$\Delta\sigma_C = 1.28\Delta\sigma'_C$$
$$\Delta\sigma_H = 1.43\Delta\sigma'_H \qquad (11)$$

In the original reports, the $\eta_C$ and $\eta_H$ rates were measured using pulse sequences based on two-dimensional $^1$H or $^{13}$C-coupled spectra[63]. In these spectra, the splitting of the multiplet components causes severe signal overlapping, leading to a decrease in the number of methyl groups that can be analysed. To overcome these difficulties, the pulse sequences were modified to separate the splitting of the multiplet in the third frequency domain. Supplementary Fig. 3 shows the pulse sequences used to measure the $\eta_C$ and $\eta_H$ rates, and the obtained spectra for measuring the $\eta_C$ rates using a {u-$^2$H, Ile$\delta$1, Leu$\delta$2, Val$\gamma$2-[$^{13}$CH$_3$]} maltose binding protein (MBP) sample. As shown in Supplementary Fig. 3c, all four multiplet components were completely resolved in the $F1$ frequency domain, enabling the comprehensive analysis of the methyl groups. It should be noted that the modified sequence for measuring the $\eta_H$ rate does not contain the purge element, which eliminates the fast-relaxing $^1$H elements[64] (Supplementary Fig. 3b). Thus, the use of the original two-dimensional based pulse sequence is recommended for smaller proteins.

The products of the order parameter and the rotational correlation time, $S^2_{\text{axis}}\tau_C$, were measured by three-quantum relaxation violated coherence transfer experiments, as established by Sun and co-workers[65,66]. The intra-methyl $^1$H–$^1$H dipolar cross-correlated relaxation rates, $\eta_{\text{HH}}$, were extracted by fitting the peak intensities measured in a pair of data sets to equation (12). $I_a(T)$ and $I_b(T)$ represent the peak intensities from two different transfer pathways with the relaxation period $T$, and $\delta$ represents the contributions from external $^1$H spins.

$$\left|\frac{I_a(T)}{I_b(T)}\right| = \frac{3}{4} \times \frac{\eta_{\text{HH}}\tanh\left(\sqrt{\eta_{\text{HH}}^2 + \delta^2}\,T\right)}{\sqrt{\eta_{\text{HH}}^2 + \delta^2} - \delta\tanh\left(\sqrt{\eta_{\text{HH}}^2 + \delta^2}\,T\right)} \tag{12}$$

The $\eta_{\text{HH}}$ can be expressed using the methyl three-fold axis order parameter, $S^2_{\text{axis}}$, the rotational correlation time, $\tau_C$, the gyromagnetic ratio, $\gamma_H$, the angle between the methyl threefold axis and a vector that connects a pair of methyl $^1$H nuclei, $\theta_{\text{axis,HH}}$, the length of the methyl $^1$H–$^1$H bond, $r_{\text{HH}}$, and $P_2(\cos x) = (3\cos^2 x - 1)/2$ (equation (13)). The product of the order parameter and the rotational correlation time, $S^2_{\text{axis}}\tau_C$, was calculated using the $\eta_{\text{HH}}$ rate.

$$\eta_{\text{HH}} = \frac{9}{10}\left(\frac{\mu_0}{4\pi}\right)^2 \left[P_2\left(\cos\theta_{\text{axis,HH}}\right)\right]^2 \frac{S^2_{\text{axis}}\gamma_H^4 \hbar^2 \tau_C}{r_{\text{HH}}^6} \tag{13}$$

**MQ relaxation analyses of MBP.** To test whether the $R_{\text{MQ,ex}}$ and $\Delta R_{\text{MQ,ex}}$ can be correctly evaluated in high molecular weight proteins, we applied the method to the MBP/$\beta$-cyclodextrin complex (MBP, molecular weight of 42 K), in which the significant chemical exchange processes were reportedly absent in the majority of the side chain methyl groups[67].

The E. coli MBP (residues 1–370) protein was expressed in E. coli BL21(DE3) cells (Agilent Technologies). For the selective $^{13}$CH$_3$-labelling of methyl groups, E. coli cells were grown in deuterated M9 media, and 50 mg l$^{-1}$ of [methyl-$^{13}$C, 3-$^2$H$_2$]-$\alpha$-ketobutyric acid (for Ile$\delta$1) (CIL) and 300 mg l$^{-1}$ of [2-methyl $^{13}$C, 4-$^2$H$_3$]-acetolactate (for Leu$\delta$2 and Val$\gamma$2) (NMR-Bio) were both added 1 h before the induction. The protein was purified sequentially with Q sepharose (GE) and Amylose Resin (New England Biolabs), from which MBP was eluted as the maltose-bound form. To exchange the bound maltose to $\beta$-cyclodextrin, MBP was denatured in 2 M guanidine hydrochloride (GuHCl). MBP was applied to a PD-10 gel filtration column (GE) to remove maltose, and refolded in GuHCl-free buffer containing 2 mM $\beta$-cyclodextrin[41,67]. The NMR sample consisted of 0.7 mM MBP, 2 mM $\beta$-cyclodextrin, 20 mM sodium phosphate buffer (pH 7.2), 3 mM NaN$_3$ and 0.1 mM EDTA in D$_2$O.

The measurements of the $R_{\text{MQ}}$ and $\Delta R_{\text{MQ}}$ rates were performed on Bruker Avance 400, 500, 600 or 800 spectrometers equipped with room temperature (400 MHz) or cryogenic (500 MHz, 600 MHz, 800 MHz) probes. The relaxation delays were varied between 3.9 and 110 ms. The measurements of the $\eta_C$ and $\eta_H$ rates were performed on a Bruker Avance 600 spectrometer equipped with a cryogenic probe, using the pulse sequences described in Supplementary Fig. 3a,b. The relaxation delays were varied between 30 and 70 ms for the measurement of the $\eta_C$ rates, and between 20 and 70 ms for the measurement of the $\eta_H$ rates. The measurements of the $\eta_{\text{HH}}$ rates were performed on a Bruker Avance 600 spectrometer equipped with a room temperature probe. The relaxation delays were varied between 3.0 and 24 ms. All NMR experiments were performed at 20 °C.

Supplementary Fig. 4a shows the plots of the $R_{\text{MQ}}$ rates of Leu160, Val293, and Ile333, as a function of the square of the $B_0$. We obtained the magnetic field-dependent variations in the $R_{\text{MQ}}$ rates, which correspond to the sum of the CSA and exchange contributions ($= R_{\text{MQ,CSA}} + R_{\text{MQ,ex}}$). Supplementary Fig. 4b shows the plots of the sum of the $R_{\text{MQ,CSA}}$ and $R_{\text{MQ,ex}}$ rates at 14.1 Tesla (600 MHz $^1$H frequency) obtained from the magnetic field-dependence (grey bars), overlaid with the plots of the $R_{\text{MQ,CSA}}$ rates calculated using the CSA values and the side-chain order parameters (orange line). The $R_{\text{MQ,ex}}$ rates at 14.1 Tesla, calculated by subtracting these two rates, are also plotted (Supplementary Fig. 4c). In the majority of the methyl groups, the field-dependent variations in the $R_{\text{MQ}}$ rates were largely attributed to the $R_{\text{MQ,csa}}$ contributions, and the $R_{\text{MQ,ex}}$ rates were smaller than 3.5 s$^{-1}$ for all methyl groups except Ile11 ($R_{\text{MQ,ex}} = 4.3$ s$^{-1}$). These results are consistent with the previous report that significant chemical exchange processes are absent[67], and thus support the proposal that the $R_{\text{MQ,ex}}$ rates are correctly obtained from the magnetic field-dependence. We assume that the slightly larger $R_{\text{MQ,ex}}$ rate for Ile11 is due to the existence of a local chemical exchange process. The existence of a chemical exchange process is also suggested from the fact that the $R_{\text{MQ,ex}}$ rate of Ile11 is larger at a lower temperature ($R_{\text{MQ,ex}} = 12$ s$^{-1}$ at 4 °C), at which the $k_{\text{ex}}$ is expected to be smaller.

In the case of MBP at 20 °C (molecular weight 42 K), the $R_{\text{MQ,CSA}}$ rates ranged from 0 to 5.7 s$^{-1}$. Therefore, the $R_{\text{MQ,CSA}}$ rates are expected to be as large as $\sim 14$ s$^{-1}$ with a molecular weight of 100 K, and about 27 s$^{-1}$ with a molecular weight of 200 K, respectively, which are comparable to the exchange contributions, $R_{\text{MQ,ex}}$ (typically on the order of $10^1$ s$^{-1}$). The exchange contributions are still expected to make large contributions to the magnetic field-dependent variations in the $R_{\text{MQ}}$ rates, and can be accurately evaluated in higher molecular weight proteins.

Thus far, the results were mainly focused on the MQ relaxation rates, $R_{\text{MQ}}$. We also point out that the same conclusions were obtained when we analysed the differential MQ relaxation rates, $\Delta R_{\text{MQ}}$ (Supplementary Fig. 4d,e).

**MQ relaxation analyses of T4 lysozyme mutant.** To validate that the $R_{\text{MQ,ex}}$ and $\Delta R_{\text{MQ,ex}}$ rates can be correctly evaluated, we applied the method to the C54T, C97A, L99A, G113A, R119P T4 lysozyme mutant (T4L, molecular weight 18 K), which is known to be in a chemical exchange process on a microsecond timescale at 20 °C (refs 68,69).

The C54T, C97A, L99A, G113A, R119P T4 lysozyme mutant (T4L, residues 1–164) was expressed in E. coli BL21 (DE3) cells (Agilent Technologies). For preparing the {u-$^2$H, Ile$\delta$1, Leu$\delta$2, Val$\gamma$2-[$^{13}$CH$_3$]} T4L sample, E. coli cells were grown in deuterated M9 media using [u-$^2$H] glucose (ISOTEC), and 50 mg l$^{-1}$ of [methyl-$^{13}$C, 3-$^2$H$_2$]-$\alpha$-ketobutyric acid (for Ile$\delta$1) (CIL) and 300 mg l$^{-1}$ of [2-methyl $^{13}$C, 4-$^2$H$_3$]-acetolactate (for Leu$\delta$2 and Val$\gamma$2) (NMR-Bio) were both added 1 h before the induction. For preparing the {u-$^2$H $^{13}$C, $^{15}$N, Ile$\delta$1-[$^{13}$CH$_3$], Leu, Val-[$^{13}$CH$_3$, $^{12}$C$^2$H$_3$]} T4L sample, E. coli cells were grown in deuterated M9 media using [u-$^2$H,$^{13}$C,$^{15}$N] glucose (CIL), and 50 mg l$^{-1}$ of [$^{13}$C$_4$, 3-$^2$H$_2$]-$\alpha$-ketobutyric acid (for Ile$\delta$1) (CIL) and 80 mg l$^{-1}$ of [1,2,3,4-$^{13}$C$_4$, 3,4',4',4'-$^2$H$_4$]-$\alpha$-ketoisovaleric acid (for Leu and Val) (CIL) were both added 1 h before the induction. The protein was purified sequentially on Hitrap SP (GE) and 26/60 Superdex 75 pg column (GE) as described previously[68,70]. The NMR sample consisted of 1.2 mM T4L, 50 mM sodium phosphate buffer (pH 5.5), 25 mM NaCl, 3 mM NaN$_3$ and 2 mM EDTA in D$_2$O.

Resonance assignments of the Ile, Leu, Val methyl signals were obtained from a set of out-and-back type triple-resonance experiments, using the {u-$^2$H $^{13}$C, $^{15}$N, Ile$\delta$1-[$^{13}$CH$_3$], Leu, Val-[$^{13}$CH$_3$, $^{12}$C$^2$H$_3$]} T4L sample. HMCM[CG]CBCA, Val-HMCM(CBCA)CO, and Ile, Leu-HMCM(CGCBCA)CO spectra were recorded at 34 °C with a Bruker Avance 500 spectrometer equipped with a cryogenic probe[71]. The assignments of the backbone chemical shifts were transferred from the previous report[68].

Before the application of the method, we performed $^1$H and $^{13}$C SQ CPMG RD analyses at 4 °C to obtain the chemical exchange parameters, using a {u-$^2$H, Ile$\delta$1, Leu$\delta$2, Val$\gamma$2-[$^{13}$CH$_3$]} T4L sample. The $^{13}$C/$^1$H SQ CPMG RD analyses of the {u-$^2$H, Ile$\delta$1, Leu$\delta$2, Val$\gamma$2-[$^{13}$CH$_3$]} T4L were recorded with Bruker Avance 600 and 800 spectrometers equipped with a room temperature (600 MHz) or a cryogenic (800 MHz) probe using the pulse sequences, which create $^{13}$C SQ or $^1$H SQ coherences during the constant-time CPMG period[72,73]. The temperature was varied between 4 °C and 25 °C. The constant-time CPMG relaxation period $T$ was set to 40 ms. Values of $R_{2,\text{eff}}$ ($\nu_{\text{CPMG}}$) were calculated using equation (14), where $I(\nu_{\text{CPMG}})$ and $I(0)$ represent the peak intensities with and without the relaxation period $T$, respectively.

$$R_{2,\text{eff}}(\nu_{\text{CPMG}}) = -\frac{1}{T}\ln\left\{\frac{I(\nu_{\text{CPMG}})}{I(0)}\right\} \tag{14}$$

The $^{13}$C/$^1$H SQ CPMG dispersion curves were fitted to the Carver–Richards formula[74] (equation (15)) with the in-house-developed program written in the programming language, Python 2.7, supplemented with the extension modules, Numpy 1.7 and Scipy 0.11.0. The equation contains four variables used to fit each dispersion curve: the intrinsic transverse relaxation rate $R_{2,0}$, the exchange rate $k_{\text{ex}}$, the chemical shift difference $\Delta\omega$ and the minor state population $p_B$. Dispersion curves collected at two static magnetic fields were fitted together. The forward and reverse rate constants at 25 °C were calculated by extrapolating the linear Arrhenius plot obtained at 4, 8, 12 and 16 °C.

$$R_{2,\text{eff}} = R_{2,0} + \frac{k_{\text{ex}}}{2} - \nu_{\text{CPMG}}\cosh^{-1}\left[D_+ \cosh(\eta_+) - D_- \cos(\eta_-)\right]$$
$$\eta_\pm = \frac{1}{2\sqrt{2}\nu_{\text{CPMG}}}\left[\pm\Psi + \left(\Psi^2 + \xi^2\right)^{1/2}\right]^{1/2}$$
$$D_\pm = \frac{1}{2}\left[\pm 1 + \frac{\Psi + 2\Delta\omega^2}{\left(\Psi^2 + \xi^2\right)^{1/2}}\right]$$
$$\Psi = k_{\text{ex}}^2 - \Delta\omega^2$$
$$\xi = -2\Delta\omega k_{\text{ex}}(1 - 2p_B)$$
(15)

All of the dispersion profiles were globally fitted with the exchange rate of $610 \pm 30$ s$^{-1}$ and the minor state population of $4.2 \pm 0.1\%$ at 4 °C, and the $^{13}$C and $^1$H chemical shift differences were obtained. To investigate the microsecond chemical exchange processes, we also performed the CPMG RD analyses at 25 °C. The obtained dispersion profiles were nearly flat, indicating that the chemical exchange process is on a microsecond timescale at a higher temperature (Supplementary Fig. 5a). We conducted the $^{13}$C SQ CPMG RD analyses and obtained the chemical exchange parameters at four different temperatures (4, 8, 12 and 16 °C; Supplementary Fig. 5b), and the exchange rate constant and the minor state population at 25 °C were estimated to be 12,000 s$^{-1}$ and 2.5%, respectively, from the extrapolated Arrhenius plot (Supplementary Fig. 5c).

We applied the magnetic field-dependent MQ relaxation analyses to T4L at 25 °C. The measurements of the $R_{\text{MQ}}$ and $\Delta R_{\text{MQ}}$ rates were performed on Bruker Avance 400, 500, 600 or 800 spectrometers equipped with room temperature (400 MHz) or cryogenic (500 MHz, 600 MHz, 800 MHz) probes. The relaxation delays were varied between 3.9 and 130 ms. The measurements of the $\eta_C$ and $\eta_H$ rates were performed on a Bruker Avance 600 spectrometer equipped with a room temperature probe, using the pulse sequences described in Supplementary Fig. 3a,b. The relaxation delays were varied between 20 and 70 ms for the measurement of the $\eta_C$ rates, and between 10 and 100 ms for the measurement of the $\eta_H$ rates. The

measurements of the $\eta_{HH}$ rates were performed on a Bruker Avance 600 spectrometer equipped with a room temperature probe. The relaxation delays were varied between 4.0 and 32 ms. All NMR experiments were performed at 25 °C.

Supplementary Fig. 6a shows the plots of the $R_{MQ}$ and $\Delta R_{MQ}$ rates of Leu133 as a function of the square of the $B_0$, and the $^{13}C$ SQ CPMG dispersion profile of Leu133 is also shown for comparison. While the change in the effective relaxation rate ($R_{2,eff}$) was relatively small up to 7 s$^{-1}$ in the CPMG RD analysis, significant magnetic field-dependent changes in the $R_{MQ}$ (16 s$^{-1}$ at 14.1 Tesla) and $\Delta R_{MQ}$ rates (30 s$^{-1}$ at 14.1 Tesla) were observed in Leu133. In Leu133, the CSA contributions to the magnetic field-dependent changes were estimated to be about 2 s$^{-1}$, and thus the observed magnetic field-dependent change largely originated from the chemical exchange contributions. Supplementary Fig. 6b shows the plots of the $R_{MQ,ex}$ and $\Delta R_{MQ,ex}$ rates at 14.1 Tesla (600 MHz $^1H$ frequency) obtained from the magnetic field-dependence. Significant chemical exchange contributions larger than 10 s$^{-1}$ were observed for Val103, Leu118, Leu133 and Ile150, which are located in the vicinity of helices F and G, consistent with the previous structural analyses[68] (Supplementary Fig. 6c). These results strongly support the proposal that the magnetic field-dependent MQ relaxation analyses are particularly beneficial for detecting chemical exchange processes on a microsecond timescale, which are difficult to detect by CPMG RD analyses.

To determine whether the sizes of the obtained $R_{MQ,ex}$ and $\Delta R_{MQ,ex}$ rates accurately reflect the chemical shift differences, we compared the observed $R_{MQ,ex}$ and $\Delta R_{MQ,ex}$ rates with the corresponding MQ chemical shift differences (($\gamma_C^2\Delta\varpi_C^2 + \gamma_H^2\Delta\varpi_H^2$) for $R_{MQ,ex}$ (equations (1)) and ($4\gamma_C\Delta\varpi_C\gamma_H\Delta\varpi_H$) for $\Delta R_{MQ,ex}$ (equation (2)), calculated using the results of the SQ CPMG RD analyses at 4 °C (Supplementary Fig. 6d). The observed $R_{MQ,ex}$ and $\Delta R_{MQ,ex}$ rates correlated well with the chemical shift differences calculated from the CPMG RD analyses, except for $\Delta R_{MQ,ex}$ of Leu118, with the slope of $3.8 \times 10^{-6}$. The slope value, corresponding to $k_{ex}/p_A p_B$, is consistent with the value of $2.0 \times 10^{-6}$, calculated using $k_{ex} = 12{,}000$ s$^{-1}$ and $p_B = 0.025$. These results support the proposal that accurate $R_{MQ,ex}$ and $\Delta R_{MQ,ex}$ rates are obtained from the magnetic field-dependent MQ relaxation analyses.

The $\Delta R_{MQ,ex}$ rate for Leu118 was nearly 0 s$^{-1}$, and apparently inconsistent with the results of the CPMG RD analyses, which showed that significant chemical shift difference exist in both $^{13}C$ and $^1H$. We assume that this is because the $^{13}C$ and $^1H$ chemical exchange processes are not correlated in Leu118. Since the $\Delta R_{MQ,ex}$ contributions originate from the differences in the exchange broadening effects between the DQ (frequency of $|\gamma_C B_0 \Delta\varpi_C + \gamma_H B_0 \Delta\varpi_H|$) and ZQ (frequency of $|\gamma_C B_0 \Delta\varpi_C - \gamma_H B_0 \Delta\varpi_H|$) coherences, the uncorrelated chemical shift modulations in $^{13}C$ and $^1H$ do not lead to deviations in the $\Delta R_{MQ,ex}$ rates. To investigate the correlation of the chemical exchange processes in $^{13}C$ and $^1H$, we compared the chemical exchange parameters obtained from the global fitting procedure with those obtained by fitting the $^{13}C$ and $^1H$ dispersion profiles separately. Supplementary Fig. 6e,f show the plots of $k_{ex}$ and $p_B$ from residues with chemical shift differences in both $^{13}C$ and $^1H$ nuclei. For Ile150 and Leu133, the $^{13}C$ and $^1H$ results were almost the same and agreed well with the results obtained from the global fitting procedure, indicating that the $^{13}C$ and $^1H$ chemical exchange processes are correlated in these methyl groups. However, in Leu118, the $^1H$ and $^{13}C$ results were different from each other and the $^1H$ results were not in agreement with the results from the global fitting procedure, indicating that the $^{13}C$ and $^1H$ chemical exchange processes are not correlated. This result is also evident from the fitting curve of the $^1H$ dispersion profile of Leu118, where the global fitting curve did not fit the observed $R_{2,eff}$ values well (Supplementary Fig. 6g). We suppose that the losses of the correlation of $^{13}C$ and $^1H$, as observed in Leu118, are due to the difference in the origin of the chemical shift changes in $^{13}C$ and $^1H$. The $^{13}C$ chemical shifts are known to reflect changes in the side chain rotamer distributions[75–79], and the $^1H$ chemical shifts reflect the changes in the configurations of the neighbouring aromatic rings[80] (Phe114 in the case of Leu118). These structural changes do not necessarily occur in a correlated manner.

### Resonance assignments of Gα-GDP.

The resonance assignments of the Alaβ, Ileδ1, Leuδ2 and Valγ2 methyl signals were performed, by combining mutagenesis and nuclear Overhauser effect (NOE) analyses, based on the crystal structure.

We constructed 11 Ala mutants (A7V, A11V, A12V, A30V, A31V, A98S, A99S, A101V, A111S, A114V and A235V), 18 Ile mutants (I19V, I49V, I55V, I56V, I162V, I184V, I212V, I221V, I222V, I253V, I264V, I265V, I278V, I285V, I303V, I319V, I343V and I344V), 18 Val mutants (V13A, V50I, V71A, V72I, V73I, V109A, V118I, V126I, V174I, V179I, V185I, V201I, V218I, V233I, V332A, V335I, V339I and V342I) and 15 Leu mutants (L5I, L23I, L36I, L37I, L38I, L39I, L110I, L175I, L232I, L234I, L249I, L273I, L310I, L348I, and L353I). All mutants were constructed with a QuikChange Site-directed Mutagenesis Kit (Agilent Technologies). We obtained the $^1H$-$^{13}C$ HMQC spectrum of each mutant and compared all of the spectra to that of the wild type. The spectra were recorded at 20 °C with Bruker Avance 500 or 600 spectrometers equipped with cryogenic probes.

To observe the methyl-backbone amide and methyl–methyl NOEs, we obtained a set of three-dimensional NOESY spectra. The [$^1H$-$^1H$] NOESY-[$^1H$-$^{15}N$] TROSY spectrum and the [$^1H$-$^1H$] NOESY-[$^1H$-$^{13}C$] HMQC spectrum were obtained for a {u-$^2H$, $^{15}N$, Alaβ, Ileδ1-[$^{13}CH_3$]} Gα-GDP sample. The [$^1H$-$^1H$] NOESY-[$^1H$-$^{13}C$] HMQC spectrum and the [$^1H$-$^{13}C$] HMQC-[$^1H$-$^1H$]

NOESY-[$^1H$-$^{13}C$] HMQC spectrum were recorded on {u-$^2H$, Alaβ, Ileδ1, Leuδ2, Valγ2-[$^{13}CH_3$]} and {u-$^2H$, Leu/Val-[$^{13}CH_3$, $^{13}CH_3$]} Gα-GDP samples, with or without the GoLoco14 peptide. The mixing time was set to 200 ms in all measurements. The spectra were recorded at 20 °C with Bruker Avance 600 or 800 spectrometers equipped with cryogenic probes. The identified NOEs were assigned using the crystal structures of Gα$_{i1}$-GDP (PDB ID: 1GDD)[17] and the complex of GoLoco14 and Gα$_{i1}$-GDP (PDB ID: 1KJY)[39].

We established 97% of the Alaβ (24/25), Ileδ1 (26/26), Leuδ2 (27/27) and Valγ2 (19/21) assignments for Gα-GDP, and 95 % of the Alaβ (23/25), Ileδ1 (23/26), Leuδ2 (27/27) and Valγ2 (21/21) assignments for GoLoco14-Gα · GDP. The results are summarized in Supplementary Fig. 2.

### CPMG RD analyses of Gα-GDP.

$^1H$-$^{13}C$ MQ/$^{13}C$ SQ CPMG RD analyses were recorded at 20 °C using the pulse sequences, which create $^1H$-$^{13}C$ MQ or $^{13}C$ SQ coherences during the constant-time CPMG periods[23,72]. The constant-time CPMG relaxation period $T$ was set to 40 ms. The values of the effective relaxation rates measured in the presence of a $\nu_{CPMG}$ Hz CPMG pulse train, $R_{2,eff}$ ($\nu_{CPMG}$), were calculated using equation (14). The uncertainties of $R_{2,eff}$ ($\nu_{CPMG}$) ($\Delta R_{2,eff}$ ($\nu_{CPMG}$)) were calculated using equation (16), where $\Delta(\nu_{CPMG})$ represents the uncertainties of $I(\nu_{CPMG})$.

$$\Delta R_{2,eff}(\nu_{CPMG}) = \frac{1}{T} \cdot \frac{1}{\Delta(\nu_{CPMG})} \qquad (16)$$

### MQ relaxation measurements of Gα-GDP.

The measurements of the $R_{MQ}$ and $\Delta R_{MQ}$ rates were performed on Bruker Avance 500, 600 or 800 spectrometers equipped with cryogenic probes. The relaxation delays were varied between 3.9 and 51 ms. The measurements of the $\eta_C$ and $\eta_H$ rates were performed on a Bruker Avance 600 spectrometer equipped with a cryogenic probe, using the pulse sequences described in Supplementary Fig. 3a,b. The relaxation delays were varied between 20 and 50 ms for the measurement of the $\eta_C$ rates, and between 10 and 40 ms for the measurement of the $\eta_H$ rates. The measurements of the $\eta_{HH}$ rates were performed on a Bruker Avance 600 spectrometer equipped with a cryogenic probe. The relaxation delays were varied between 2.0 and 12 ms. All NMR experiments were performed at 20 °C.

### Code availability.

Codes for curve fitting and error analysis are available from the corresponding author upon reasonable request.

### Data availability.

Sequence information on human Gα$_{i3}$, rat RGS14, E.coli MBP, bacteriophage T4 lysozyme are available in the UniProt Knowledgebase under accession codes P08754, O08773, P0AEX9, and P00720. The PDB accession codes 1GDD, 1KJY, 1DMB and 3DMV were used in this study. The backbone and methyl resonance assignments for Gα-GDP and the methyl resonance assignments for GoLoco14 · Gα have been deposited in the BMRB with the accession numbers 19015, 26975 and 26976. All other data are available from the corresponding author upon reasonable request.

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

## Acknowledgements

This work was supported in part by grants from the Japan New Energy and Industrial Technology Development Organization (NEDO) and the Ministry of Economy, Trade and Industry (METI) (to I.S.), a Grant-in-Aid for Scientific Research on Priority Areas from the Japanese Ministry of Education, Culture, Sports, Science and Technology (MEXT) (to I.S.), the Development of core technologies for innovative drug development based upon IT from Japan Agency for Medical Research and development, AMED (to I.S.), JSPS KAKENHI Grant Number JP16H01368 (to M.O.), a grant from The Vehicle Racing Commemorative Foundation (to M.O.), and a grant from Nagase Science Technology Foundation (to M.O.).

## Author contributions

Y.T., H.K., Y.M., M.Y., M.O. and I.S. designed the study, Y.T., H.K., Y.M. and M.Y. performed the experiments and Y.T., H.K., Y.M., M.Y., M.O. and I.S. analysed the data and wrote the paper.

## Additional information

**Competing financial interests:** The authors declare no competing financial interests.

