## [Peer review file · Nature Communications]

Reviewers' comments:

Reviewer #1 (Remarks to the Author):

The manuscript describes the characterization of dynamic processes in the subunit alpha of a G protein in states of faster and slower GDP dissociation. Dynamic processes in G proteins are likely to be of high importance for signal transduction initiated by ligand-binding to GPCRs. To dissect the dynamic processes the authors develop a new NMR approach based on multiple-quantum relaxation in methyl groups. After demonstration that the new experiments are robust (using test systems of MBP and T4 lysozyme, which results in an extensive Supplementary Material), they apply these experiments to Galpha protein with the wild-type sequence as well as a known mutation (at residue 150), for which faster GDP-dissociation is demonstrated, and a complex with a peptide. Using these experiments the authors convincingly demonstrate that changes in conformational exchange processes affecting the methyl groups are connected to GDP dissociation.

I would like to congratulate the authors to this fantastic piece of research. It is clear that the NMR experiments were very well designed and provide unique insights into functional dynamics of a G protein in different, biologically relevant states. I am also convinced that the proposed experiments will be used in the future for a wide range of applications, which utilize methyl-labeled high-molecular weight proteins.

I just have very few minor suggestions for improvement:

-On page 6 it is stated that “structure of wild type 101 Gα·GDP17 revealed that Asp150, located on the αE helix in the helical domain, forms no direct interactions with the bound GDP”. From the current figures this is not so clear. Indeed, Fig. S1b suggests that the side-chain of Asp150 is very close to the bound GDP.

-On page 9 it is stated that “Assuming a slow tumbling limit and a fast exchange regime (exchange rates, $k_{ex} \gg$ chemical shift differences, $\Delta\omega$), which usually hold for the microsecond exchange processes in proteins,”. I am not sure that such a general statement is justified. This should be more clearly specified.

-Fig. 2b and Supp. Mat Fig.: Please include error bars in plots showing relaxation rates as a function of magnetic field strength.

Reviewer #2 (Remarks to the Author):

In their manuscript Toyama et al. report the dynamics of wildtype G α i3 and an oncogenic mutant G α i3(D150N) in its GDP bound (inactive) form. They can identify two states of the protein, one state with a reduced affinity for GDP that probably is involved in the GDP release. In the oncogenic mutant the GDP-exchange is 20-fold faster than in the wildtype. The magnetic field-dependent NMR relaxation analysis reveals that in the mutant the dynamics between the two states is changed and the population of the excited conformation is increased. Binding of a peptide from RGS14 protein reduces the dissociation of GDP by modulating the conformational equilibrium processes within G α . In addition, they present a method to determine the dynamics of methyl groups in proteins where the methyl groups are ¹³C-labelled and test this method with two different proteins.

The paper has its merits but in the present form I would not recommend it for publication in Nature Communication.

Specific remarks:

1. The authors describe in the paper the dynamics of G α that are interesting from the biological point of view and mix it with a rather lengthy representation of an improved extraction of exchange rates from MQ relaxation dispersion measurements. Additionally, they describe a test of their method with two proteins. In my opinion this part should be published in a typical NMR journal such as the J. Biomol. NMR or be shifted completely to the supplement.

2. In an excellent recent paper (cited by the authors), Goricanec et al describe the dynamics of G α i1 in the absence of nucleotide, with GDP and GTP and GTP analogues bound. These data are also measured by amide and methyl relaxation dispersion. In addition, they give structural details of the two states and describe the effect of GPCR binding. That means that the equilibrium they observe has already been described in a related G α i. The influence of an oncogenic mutant on this equilibrium is new and fits to the model (that would correspond to a state 1 to state 2 transition in Ras). In addition they describe the interaction with the interaction peptide of the GAP activator RGS14, determine GDP exchange rates and the interaction surface. All these data are of biological importance.

3. An interesting observation is that the concentration of free Mg²⁺ influences strongly the exchange rates. This explains the large discrepancies between their exchange rates and those given by Goricanec et al. The equilibrium as well as the exchange rates are strongly dependent on the Mg²⁺ concentration. This is an important observation that should be presented in the main part of the paper. A possible regulatory influence of magnesium on G-proteins has been discussed in literature for a long time. Here, they can simply see it in the NMR spectra.

4. "The ¹³CH₂H₂ labeling requires costly precursors (typically 3 to 4 fold more expensive), as compared to the conventional ¹³CH₃ labeling" this is true, but the costs of the precursor are only a small part of the cost of the complete medium. Making a sample with CH₃ or CHD₂ is not so much different in price.

5. I do not understand why they can have negative relaxation rates (Fig. 3). The authors should comment this.

Reviewer 1

The manuscript describes the characterization of dynamic processes in the subunit alpha of a G protein in states of faster and slower GDP dissociation. Dynamic processes in G proteins are likely to be of high importance for signal transduction initiated by ligand-binding to GPCRs. To dissect the dynamic processes the authors develop a new NMR approach based on multiple-quantum relaxation in methyl groups. After demonstration that the new experiments are robust (using test systems of MBP and T4 lysozyme, which results in an extensive Supplementary Material), they apply these experiments to G α protein with the wild-type sequence as well as a known mutation (at residue 150), for which faster GDP-dissociation is demonstrated, and a complex with a peptide. Using these experiments the authors convincingly demonstrate that changes in conformational exchange processes affecting the methyl groups are connected to GDP dissociation. I would like to congratulate the authors to this fantastic piece of research. It is clear that the NMR experiments were very well designed and provide unique insights into functional dynamics of a G protein in different, biologically relevant states. I am also convinced that the proposed experiments will be used in the future for a wide range of applications, which utilize methyl-labeled high-molecular weight proteins.

I just have very few minor suggestions for improvement:

1. On page 6 it is stated that “structure of wild type 101 G α :GDP17 revealed that Asp150, located on the α E helix in the helical domain, forms no direct interactions with the bound GDP”. From the current figures this is not so clear. Indeed, Fig. S1b suggests that the side-chain of Asp150 is very close to the bound GDP.

I agree with the reviewer's comment. In the revised manuscript, we added an additional enlarged view of the GDP-binding region from another point of view, in Supplementary Figure 1b (highlighted in yellow), so that readers can clearly see the position of Asp 150.

(Page S2, Supplementary Figure 2b)

(Revised) Supplementary Figure 1 Structure of $G\alpha$ -GDP

(a) Crystal structure of $G\alpha$ -GDP (PDB ID : 1GDD)¹. The switch regions are colored green. The regions lacking the electron density are depicted by dashed lines. (b) An enlarged view of the GDP-binding region. The phosphate binding loop (P-loop) is colored magenta. (c) A schematic representation of the $G\alpha$ -GDP interactions.

2. On page 9 it is stated that “Assuming a slow tumbling limit and a fast exchange regime (exchange rates, $k_{ex} \gg$ chemical shift differences, $\Delta\omega$), which usually hold for the microsecond exchange processes in proteins.”. I am not sure that such a general statement is justified. This should be more clearly specified.

I appreciate the reviewer’s comment. In the revised manuscript, we added an explanation regarding this point as follows.

(Page S15, line 135 -page S16 line 146)

Here, we assume a slow tumbling limit ($\omega^2\tau_c^2 \gg 1$, where ω represents Larmor frequency and τ_c represents rotational correlation time) and a fast exchange regime (exchange rates, $k_{ex} \gg$ chemical shift differences, $\Delta\omega$). In the case of proteins with molecular weights larger than 20 K (corresponding to the correlation time of ~ 12 ns), the value of $\omega^2\tau_c^2$ is estimated to be larger than 10^2 . In addition, the MQ chemical shift differences in $^{13}\text{CH}_3$ methyl groups are usually smaller than 500 Hz at 14.1 Tesla, as judged from the case of T4 lysozyme mutant (see Supplementary Method 3), which is below the exchange rates of the microsecond exchange processes with the order of $10^3 \sim 10^5 \text{ s}^{-1}$. Therefore, the assumptions of a slow tumbling limit and a fast exchange regime usually hold for the microsecond exchange processes in high molecular weight proteins. Under these assumptions, the CSA and exchange contributions are both proportional to the square of the static magnetic field strength, B_0 .

3. Fig. 2b and Supp. Mat Fig.: Please include error bars in plots showing relaxation rates as a function of magnetic field strength.

I appreciate the reviewer’s comment. Since the experiments for measuring R_{MQ} rates are very sensitive and a sufficient number of time points (> 6) are sampled, the errors for the obtained R_{MQ} rates are very small (typically smaller than 2 s^{-1}). Although it is hard to see the error bars in the plots, we added the error bars in the plots of the R_{MQ} rates as a function of B_0 in the revised manuscript.

(Figure 2b)

(Revised) **Figure 2 Magnetic field-dependent MQ relaxation analyses of the wild type $G\alpha \cdot \text{GDP}$** (a) Schematic representation of the magnetic field-dependent MQ relaxation analyses. (b) Plots of the R_{MQ} rates of Leu38, Leu39, Ile85, and Ile253 against the square of the static magnetic fields. The obtained $R_{MQ,ex}$ rates at 14.1 Tesla (600 MHz ^1H frequency) are shown. (c) Plots of the $R_{MQ,ex}$ (top) and $\Delta R_{MQ,ex}$ (bottom) rates at 14.1 Tesla. The methyl groups with $R_{MQ,ex}$ or $\Delta R_{MQ,ex}$ rates larger than 10 s^{-1} are colored magenta, and those within the 5.0 to 10 s^{-1} range are colored pink. The small negative values of the $R_{MQ,ex}$ rates are due to the overestimations of the CSA contributions to the MQ relaxation rates, originating from

systematic errors in the measurements. We suppose that the relatively large negative value of the $R_{MQ,ex}$ rate of Val332 is due to its low signal-to-noise ratio and its very large dipolar contribution to the MQ relaxation rate (44 s^{-1}), which hampers the reliable measurement of the ^1H - ^{13}C cross correlated relaxation rate (1.9 s^{-1} on average). (d) Mapping of the methyl groups with significant $R_{MQ,ex}$ or $\Delta R_{MQ,ex}$ rates on the structure of Gα·GDP (PDB ID: 1GDD)¹⁷. The methyl groups with significant $R_{MQ,ex}$ or $\Delta R_{MQ,ex}$ rates are colored in the same manner as in (c). Methyl groups with no data are colored gray.

(Page S7, Supplementary Figure S4a)

(Revised) Supplementary Figure 4 Magnetic field-dependent MQ relaxation analyses of MBP (a) Plots of the R_{MQ} rates of Leu160, Val293, and Ile333 against the square of the static magnetic fields. The sum of the $R_{MQ,CSA}$ and $R_{MQ,ex}$ rates of Val293 at 14.1 Tesla (600 MHz 1H frequency) is shown. (b) Plots of the sum of the $R_{MQ,CSA}$ and $R_{MQ,ex}$ rates (gray bars) and the $R_{MQ,CSA}$ rates (orange line). (c) Plot of the $R_{MQ,ex}$ rates. (d) Plots of the sum of the $\Delta R_{MQ,CSA}$ and $\Delta R_{MQ,ex}$ rates (gray bars) and the $\Delta R_{MQ,CSA}$ rates (orange line). (e) Plot of the $\Delta R_{MQ,ex}$ rates. The results from the Ile δ 1, Leu δ 2, Val γ 2 methyl groups in MBP at 14.1 Tesla (600 MHz 1H frequency) are shown.

(Page S10, Supplementary Figure S6a)

(Revised) Supplementary Figure 6 Magnetic field-dependent MQ relaxation analyses of

T4L (a) The SQ CPMG RD profile (left) and the plot of the R_{MQ} (middle) and ΔR_{MQ} (right) rates against the square of the static magnetic field of Leu133. The change in the effective relaxation rates in the SQ CPMG RD profile is shown, and the chemical exchange contributions to the MQ relaxation rates at 14.1 Tesla (600 MHz 1H frequency) are shown. The experiments were performed at 298 K. (b) Plots of the $R_{MQ,ex}$ (top) and $\Delta R_{MQ,ex}$ (bottom) rates of the Ile δ 1, Leu δ 2, and Val γ 2 methyl groups of T4L. The methyl groups with significant chemical exchange contributions larger than 10 s^{-1} are colored magenta. (c) Mapping of the methyl groups with significant chemical exchange contributions on the crystal structure of the T4L mutant (PDB ID: 3DMV)⁴. Methyl groups with chemical exchange contributions larger than 10 s^{-1} are colored magenta. (d) Linear correlation plots of the $R_{MQ,ex}$ (left) and $\Delta R_{MQ,ex}$ (right) rates obtained from the magnetic field-dependence, against the MQ chemical shift differences, calculated using the results of the CPMG RD experiments. (e, f) Plots of the k_{ex} (e) and p_B (f) values obtained by fitting the ^{13}C and 1H dispersion profiles separately. The values obtained from the global fitting procedure are shown. (g) Fitting curves of the 1H dispersion profile of Leu118. The fitting curve obtained from the global fitted k_{ex} and p_B values is shown as a black line, and that obtained from the residue-specific k_{ex} and p_B values is shown as a red line.

Reviewer 2

In their manuscript Toyama et al. report the dynamics of wildtype Gai3 and an oncogenic mutant Gai3 (D150N) in its GDP bound (inactive) form. They can identify two states of the protein, one state with a reduced affinity for GDP that probably is involved in the GDP release. In the oncogenic mutant the GDP-exchange is 20-fold faster than in the wildtype. The magnetic field-dependent NMR relaxation analysis reveals that in the mutant the dynamics between the two states is changed and the population of the excited conformation is increased. Binding of a peptide from RGS14 protein reduces the dissociation of GDP by modulating the conformational equilibrium processes within Ga. In addition, they present a method to determine the dynamics of methyl groups in proteins where the methyl groups are ^{13}C -labelled and test this method with two different proteins.

The paper has its merits but in the present form I would not recommend it for publication in Nature Communication.

Specific remarks:

1. The authors describe in the paper the dynamics of Ga that are interesting from the

biological point of view and mix it with a rather lengthy representation of an improved extraction of exchange rates from MQ relaxation dispersion measurements. Additionally, they describe a test of their method with two proteins. In my opinion this part should be published in a typical NMR journal such as the J. Biomol. NMR or be shifted completely to the supplement.

I agree with the reviewer's comment. In the revised manuscript, we deleted the section '**Establishment of magnetic field-dependent multiple quantum relaxation analyses to characterize the conformational exchange processes on a microsecond time scale**', and moved a large part of the description of the method to the Supplementary Method 1. We simplified the explanations of the method, as follows.

(Page 7 line 144-page 9 line 175)

MQ relaxation analyses of G α :GDP

Although the MQ CPMG RD experiments are beneficial for characterizing the chemical exchange processes in high molecular weight proteins, the accessible time scales are often limited to exchange rates of $10^2 \sim 10^3 \text{ s}^{-1}$, mainly due to the upper limit of the frequencies of the CPMG pulse trains. The faster exchange processes, with exchange rates on the order of $10^3 \sim 10^4 \text{ s}^{-1}$, can be characterized by spin-lock based $R_{1\rho}$ dispersion experiments²⁴. However, when observing methyl groups, the $R_{1\rho}$ dispersion experiments require $^{13}\text{CH}_2\text{H}_2$ labeled samples, in order to avoid artifacts derived from intra-methyl dipolar cross correlation²⁵. The $^{13}\text{CH}_2\text{H}_2$ labeling results in lower sensitivities (about 3-fold), mainly due to the reduced initial magnetization²⁶. Therefore, the applications of the $R_{1\rho}$ dispersion experiments have been limited to small or medium sized proteins, so far^{25,27}, and a new strategy is needed for characterizing the microsecond order conformational exchange processes in high molecular weight proteins, such as G α :GDP.

In order to overcome these difficulties, we developed an NMR method to detect and characterize the chemical exchange processes on a microsecond time scale from the side-chain methyl ^1H - ^{13}C multiple quantum (MQ) relaxation rates, by utilizing the static magnetic field dependency²⁸⁻³⁰. Assuming a simple two-state (states A and B) chemical exchange process, the exchange contributions in the ^1H - ^{13}C MQ relaxation rates, $R_{\text{MQ,ex}}$ and $\Delta R_{\text{MQ,ex}}$, can be expressed by equations (1) and (2), using the exchange rate, k_{ex} , the populations of the two states, p_A , p_B ($p_A > p_B$), and the ^{13}C and ^1H chemical shift differences given in ppm units, $\Delta\varpi_{\text{C}}$ and $\Delta\varpi_{\text{H}}$ ^{31,32}.

$$R_{MQ,ex} = \frac{(\gamma_C^2 \Delta\omega_C^2 + \gamma_H^2 \Delta\omega_H^2) p_A p_B}{k_{ex}} \cdot B_0^2 \quad (1)$$

$$\Delta R_{MQ,ex} = \frac{4\gamma_C \Delta\omega_C \gamma_H \Delta\omega_H p_A p_B}{k_{ex}} \cdot B_0^2 \quad (2)$$

The $R_{MQ,ex}$ and $\Delta R_{MQ,ex}$ rates can be extracted, using (i) the R_{MQ} and ΔR_{MQ} rates measured at different static magnetic fields, (ii) the CSA values, $\Delta\sigma_C$ and $\Delta\sigma_H$, and (iii) the product of the order parameter and the rotational correlation time, $S_{axis}^2 \tau_C$, as summarized in Fig. 2a. We verified that the $R_{MQ,ex}$ and $\Delta R_{MQ,ex}$ rates are highly sensitive to the chemical exchange processes on a microsecond time scale, and that accurate $R_{MQ,ex}$ and $\Delta R_{MQ,ex}$ rates can be obtained by the magnetic field-dependent MQ relaxation analyses in high molecular weight proteins (Supplementary Methods 1-4, Supplementary Fig. 3-6).

2-1. In an excellent recent paper (cited by the authors), Goricanec et al describe the dynamics of Gai1 in the absence of nucleotide, with GDP and GTP and GTP analogues bound. These data area also measured by amide and methyl relaxation dispersion. In addition, they give structural details of the two states and describe the effect of GPCR binding.

I appreciate the reviewer's comment. As the reviewer pointed out, the elegant work by Goricanec described the structural and dynamic properties of Gai1 in the absence of nucleotide, with GDP, and with a GTP analogue, and clearly provided valuable insights into the GPCR-catalyzed activation mechanism of G α . However, the CPMG relaxation dispersion analyses by Goricanec mainly focused on methyl groups, and only a few amide residues were analyzed in the work. From this point of view, we emphasize that our MQ relaxation analyses cover all of the methyl groups with nearly complete assignments (97 % assignments), and the comprehensiveness of our analysis is nearly equal to that of the Goricanec's work (~ 85 % assignments as judged from Fig. S6 in Ref. 1).

2-2. That means that the equilibrium they observe has already described in a related Gai.

We consider that the equilibrium revealed by Goricanec to be distinct from that presented in our manuscript, because the timescales are very different. The exchange processes revealed by Goricanec are on a millisecond time scale, and those in our work are on a microsecond time scale. In addition, their millisecond time scale exchange processes are observed only in the presence of a high Mg²⁺ concentration (5 mM). Therefore, these exchange processes are probably related to different functions of G α . Importantly, our

results were obtained under physiologically relevant conditions, because the majority of $G\alpha \cdot GDP$ (> 70 %) is estimated to be in the Mg^{2+} -unbound state at the physiological Mg^{2+} concentration (within the range of 0.2 ~ 1.2 mM (Ref 2, 3)) . Therefore, we believe our results still provide dynamic insight into the physiological functions of $G\alpha$. In order to clarify these points, we added an explanation regarding this point in the revised manuscript, as follows.

(Page 12, lines 241-246)

We also confirmed that the magnetic field-dependent changes in the R_{MQ} and ΔR_{MQ} rates were still observed in the presence of Mg^{2+} . These results support our proposal that the millisecond exchange processes observed in the presence of Mg^{2+} are different exchange processes from those observed in the magnetic field-dependent MQ relaxation analyses, and these distinct exchange processes are probably related to the different functions of $G\alpha$.

2-3. The influence of an oncogenic mutant on this equilibrium is new and fits to the model (that would correspond to a state 1 to state 2 transition in Ras). In addition they describe the interaction with the interaction peptide of the GAP activator RGS14, determine GDP exchange rates and the interaction surface. All these data are from biological importance.

I appreciate the reviewer's thoughtful suggestion. In our analyses, conformational exchange processes are observed in the switch regions of $G\alpha$, as shown in Figure 2. The studies of Ras revealed that Ras is exchanging between state 1 and state 2, which show structural differences in switch 1 and switch 2. Given these similarities and the high structural homology between $G\alpha$ and Ras, the conformational exchange processes observed in $G\alpha$ might correspond to a state 1 to state 2 transition in Ras.

In the revised manuscript, we added a discussion about the relevance to the state 1 to state 2 transition in Ras, as follows.

(Page 18, lines 388-395)

Our NMR results also indicate that the conformational exchange processes exist in the switch regions. Studies of the small GTPase Ras have revealed that the GTP-bound Ras is exchanging between different conformational states, called state 1 and state 2, which show structural differences in switch 1 and switch 2, and that the transitions between these two states are closely related to the guanine nucleotide association/dissociation kinetics⁴⁷⁻⁴⁹. Given the high structural homology between the $G\alpha$ GTPase domain and Ras, the

conformational exchange processes observed in G α might correspond to the state 1 to state 2 transition in Ras.

(Page 27, line 496 - page 28 line 603)

47. Geyer, M. *et al.* Conformational Transitions in p21 ras and in Its Complexes with the Effector Protein Raf-RBD and the GTPase Activating Protein GAP. *Biochemistry* **35**, 10308–10320 (1996).

48. Liao, J. *et al.* Two conformational states of Ras GTPase exhibit differential GTP-binding kinetics. *Biochem. Biophys. Res. Commun.* **369**, 327–332 (2008).

49. Matsumoto, S. *et al.* Molecular Mechanism for Conformational Dynamics of Ras·GTP Elucidated from In-Situ Structural Transition in Crystal. *Sci. Rep.* **6**, 25931 (2016).

3. An interesting observation is that the concentration of free Mg²⁺ influences strongly the exchange rates. This explains the large discrepancies between their exchange rates and those given by Goricanec et al. The equilibrium as well as the exchange rates are strongly dependent on the Mg²⁺ concentration. This is an important observation that should be presented in the main part of the paper. A possibly regulatory influence of magnesium on G-proteins has discussed in literature for a long time. Here, they can simply see it in the NMR spectra.

I agree with the reviewer's comment. In the revised manuscript, we moved the supplementary result "***The effect of Mg²⁺ on GaGDP***" and Supplementary Figure S8 to the main text.

4. "The ¹³CHD₂ labeling requires costly precursors (typically 3 to 4 fold more expensive), as compared to the conventional ¹³CH₃ labeling" this is true, but the costs of the precursor are only a small part of the cost of the complete medium. Making a sample with CH₃ or CHD₂ is not so much different in price.

I agree with the reviewer's comment. We compared the costs of the complete medium for ¹³CH₃-labeling and that for ¹³CHD₂-labeling, using prices from Cambridge Isotope Laboratories (<https://www.isotope.com/>) (See table below). When we prepare a sample in which Ile, Leu, and Val residues are isotopically labeled, it costs \$ 1,659 for 1 L of ¹³CH₃-labeling medium, and \$ 2,023 for 1 L of ¹³CHD₂-labeling medium. Therefore, the ¹³CHD₂-labeling is only 1.2-fold more expensive than the ¹³CH₃-labeling, and the sentence "***The ¹³CHD₂ labeling requires costly precursors***" is misleading. In the revised manuscript, we deleted the sentence "***The ¹³CHD₂ labeling requires costly precursors***

(typically 3 to 4 fold more expensive), as compared to the conventional $^{13}\text{CH}_3$ labeling".

Table. Prices of the methyl-labeling precursors (<http://www.isotope.com/>)

D ₂ O 1 L		\$ 880
² H-glucose 2 g		\$ 640
¹³ CHD ₂ labeling	α-ketoisovaleric acid, (3-methyl- ¹³ C, 99%; 3-methyl-D2,3,4,4,4,D4, 98%), 80 mg	\$ 253
	α-ketobutyric acid, (4- ¹³ C, 99%; 3,3,4,4-D4, 98%), 50 mg	\$ 250
¹³ CH ₃ labeling	α-ketoisovaleric acid, (3-methyl- ¹³ C, 99%; 3,4,4,4,D4, 98%), 80 mg	\$ 80
	α-ketobutyric acid, (methyl- ¹³ C, 99%; 3,3-D2, 98%), 50 mg	\$ 59

(Page 7, line 152 - page 8 157)

Although the MQ CPMG RD experiments are beneficial for characterizing the chemical exchange processes in high molecular weight proteins, the accessible time scales are often limited to exchange rates of $10^2 \sim 10^3 \text{ s}^{-1}$, mainly due to the upper limit of the frequencies of the CPMG pulse trains. The faster exchange processes, with exchange rates on the order of $10^3 \sim 10^4 \text{ s}^{-1}$, can be characterized by spin-lock based R_{Irho} dispersion experiments²⁴. However, when observing methyl groups, the R_{Irho} dispersion experiments require $^{13}\text{CH}^2\text{H}_2$ labeled samples, in order to avoid artifacts derived from intra-methyl dipolar cross correlation²⁵. The $^{13}\text{CH}^2\text{H}_2$ labeling requires costly precursors (typically 3 to 4 fold more expensive), as compared to the conventional $^{13}\text{CH}_3$ labeling, and results in lower sensitivities (about 3-fold), mainly due to the reduced initial magnetization²⁶. Therefore, the applications of the R_{Irho} dispersion experiments have been limited to small or medium sized proteins, so far^{25,27}, and a new strategy is needed for characterizing the microsecond order conformational exchange processes in high molecular weight proteins, such as $\text{G}\alpha\cdot\text{GDP}$.

5. I do not understand why they can have negative relaxation rates (Fig. 3). The authors should comment this.

I appreciate the reviewer's comment. We suppose that the small negative values of the $R_{\text{MQ,ex}}$ rates are due to the overestimations of the CSA contributions to the MQ relaxation rates, originating from systematic errors in the CSA values and $S_{\text{axis}}^2 T_C$.

Especially, the relatively large negative value of the $R_{MQ,ex}$ rate of Val332 is probably due to its low signal-to-noise ratio and its very large dipolar contribution to the MQ relaxation rate (44 s^{-1}), which hampers the reliable measurement of the ^1H - ^{13}C cross correlated relaxation rate (1.9 s^{-1} on average). In the revised manuscript, we added an explanation regarding these points, as follows.

(Page 31, line 665 - page 32 line 681)

Figure 2 Magnetic field-dependent MQ relaxation analyses of the wild type $G\alpha\cdot\text{GDP}$

(a) Schematic representation of the magnetic field-dependent MQ relaxation analyses. (b) Plots of the R_{MQ} rates of Leu38, Leu39, Ile85, and Ile253 against the square of the static magnetic fields. The obtained $R_{MQ,ex}$ rates at 14.1 Tesla (600 MHz ^1H frequency) are shown. (c) Plots of the $R_{MQ,ex}$ (top) and $\Delta R_{MQ,ex}$ (bottom) rates at 14.1 Tesla. The methyl groups with $R_{MQ,ex}$ or $\Delta R_{MQ,ex}$ rates larger than 10 s^{-1} are colored magenta, and those within the 5.0 to 10 s^{-1} range are colored pink. The small negative values of the $R_{MQ,ex}$ rates are due to the overestimations of the CSA contributions to the MQ relaxation rates, originating from systematic errors in the measurements. We suppose that the relatively large negative value of the $R_{MQ,ex}$ rate of Val332 is due to its low signal-to-noise ratio and its very large dipolar contribution to the MQ relaxation rate (44 s^{-1}), which hampers the reliable measurement of the ^1H - ^{13}C cross correlated relaxation rate (1.9 s^{-1} on average). (d) Mapping of the methyl groups with significant $R_{MQ,ex}$ or $\Delta R_{MQ,ex}$ rates on the structure of $G\alpha\cdot\text{GDP}$ (PDB ID: 1GDD)¹⁷. The methyl groups with significant $R_{MQ,ex}$ or $\Delta R_{MQ,ex}$ rates are colored in the same manner as in (c). Methyl groups with no data are colored gray.

References

1. Goricanec, D. et al. Conformational dynamics of a G-protein α subunit is tightly regulated by nucleotide binding. *Proc. Natl. Acad. Sci. USA* **113**, E3629–E3638 (2016).
2. Levy, L. a, Murphy, E., Raju, B. & London, R. E. Measurement of cytosolic free magnesium ion concentration by ^{19}F NMR. *Biochemistry* **27**, 4041–8 (1988).
3. London, R. E. Methods for Measurement of Intracellular Magnesium:NMR and Fluorescence. *Annu. Rev. Physiol.* **53**, 241–258 (1991).

REVIEWERS' COMMENTS:

Reviewer #1 (Remarks to the Author):

As indicated in my first review, I think this is a very elegant study. In the revised version of the manuscript, the authors have addressed my minor comments.

Reviewer #2 (Remarks to the Author):

I am satisfied with the modified manuscript and recommend its publication.